# Morphing Tokens Draw Strong Masked Image Models

**Taekyung Kim*, Byeongho Heo, Dongyoon Han***
NAVER AI Lab
{taekyung.k, bh.heo, dongyoon.han}@navercorp.com

## Abstract

Masked image modeling (MIM) has emerged as a promising approach for pre-training Vision Transformers (ViTs). MIMs predict masked tokens token-wise to recover target signals that are tokenized from images or generated by pre-trained models like vision-language models. While using tokenizers or pre-trained models is viable, they often offer spatially inconsistent supervision even for neighboring tokens, hindering models from learning discriminative representations. Our pilot study identifies spatial inconsistency in supervisory signals and suggests that addressing it can improve representation learning. Building upon this insight, we introduce Dynamic Token Morphing (DTM), a novel method that dynamically aggregates tokens while preserving context to generate contextualized targets, thereby likely reducing spatial inconsistency. DTM is compatible with various SSL frameworks; we showcase significantly improved MIM results, barely introducing extra training costs. Our method facilitates MIM training by using more spatially consistent targets, resulting in improved training trends as evidenced by lower losses. Experiments on ImageNet-1K and ADE20K demonstrate DTM's superiority, which surpasses complex state-of-the-art MIM methods. Furthermore, the evaluation of transfer learning on downstream tasks like iNaturalist, along with extensive empirical studies, supports DTM's effectiveness. Code is available at https://github.com/naver-ai/dtm.

## 1 Introduction

Since the success of Vision Transformers (ViTs) (Dosovitskiy et al., 2021), numerous training strategies developed for ViTs, including self-supervised learning (SSL) methods (Chen et al., 2020; He et al., 2019; Grill et al., 2020; Caron et al., 2021). Recent advances in masked image modeling (MIM) (Zhou et al., 2022; He et al., 2022; Peng et al., 2022; Baevski et al., 2022; Heo et al., 2025; Kim et al., 2024) solidified its position as a primary SSL approach for ViT. The crux of the MIM methods is leveraging token-wise optimization objectives by predicting masked tokens to match given targets. MIM methods explored various approaches to assign effective target tokens, employing supervision from various pre-trained models, including vision-language models (Bao et al., 2022; Peng et al., 2022), utilizing momentum encoders (Baevski et al., 2022; Zhou et al., 2022), or directly exploiting patchified images (He et al., 2022; Xie et al., 2022).

While tokenizers or pre-trained models have proven effective supervisory signals for MIM targets (Bao et al., 2022; Peng et al., 2022; Li et al., 2022b; Wei et al., 2022c), we argue they often generate spatially noisy token representations with respect to class information (*i.e.*, inconsistent class labels across tokens), which may impede training when utilized as pre-training targets. For example, a pre-trained vision-language model exhibits spatially inconsistent per-token prediction results, as shown in Fig. 1. To explore the spatial inconsistency in token representations from pre-trained models, we analyze its effects on the models' capability. Our pilot exploration shows low accuracy metrics such as zero-shot classification without token aggregation (*i.e.*, denoising). Our subsequent pilot study on representation learning shows that using supervisory signals alone or signals after token aggregation without preserving context both disrupt pre-training. Since token-wise objectives are standard, spatially inconsistent targets likely challenge learning one-to-one mapping.

---

*Equal contribution

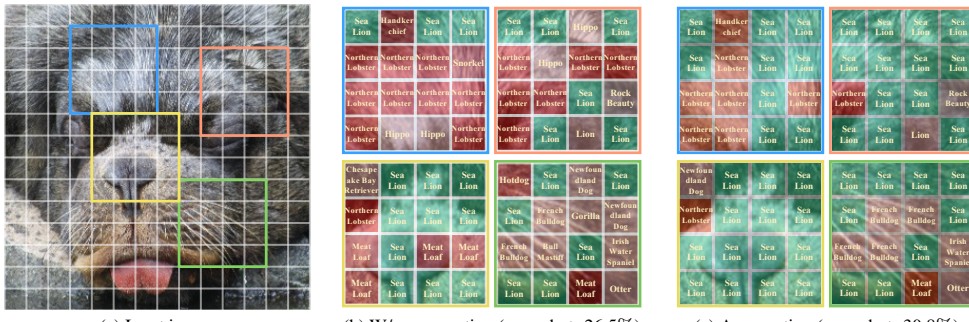

| (a) Input image | (b) W/o aggregation (zero-shot: 26.5%) | (c) Aggregation (zero-shot: 30.8%) |

Figure 1: **What is *spatial consistency* among visual tokens?** We schematically visualize *token-wise zero-shot classification* results to illustrate the spatially inconsistent token predictions. With the input image (a), the following results (b) and (c) display the predicted classes for each token within four example bounding boxes without/with token aggregations, respectively. We depict the differences between the predicted and ground-truth classes by varying the lightness of red, whereas the green represents the correct prediction. Each result yields 113 corrected tokens with aggregation and 82 without aggregation out of a total of 196 tokens, respectively; aggregation gives fewer spatially inconsistent representations. The zero-shot accuracies (reported in Table 1) support spatial consistency's connection to the model's ability.

Bearing this in mind, we introduce a novel token contextualization method called Dynamic Token Morphing (DTM), where *token morphing* aggregates related tokens while preserving context to produce coherent representations for the supervisory signal. We conjecture that training can be accelerated through the guidance of composite representations of morphed tokens derived from the preserved context even after token aggregations. Specifically, we encode the token-wise target representations and derive matching relations among tokens using DTM. The token merging process is applied to both online and target tokens considering their matching relation; it aligns each morphed token with the corresponding morphed target token while preserving the number of original tokens. The range of morphing can vary from a single token to all tokens, covering from token-wise to image-level representation learning. Among various options, we opt for bipartite matching for morphing, achieving both efficiency and efficacy.

Through extensive experiments, we verify our method's general applicability and scalability. DTM could improve fine-tuning accuracies on ImageNet (Russakovsky et al., 2015) and ADE20K (Zhou et al., 2017), which achieve state-of-the-art results. The effectiveness of our method is supported by accelerated fine-tuning trends after DTM pre-training, which highlights how spatially consistent targets are crucial. Our method shows further transferability on the iNaturalist (Van Horn et al., 2018) and fine-grained visual classification datasets (Van Horn et al., 2015; Krizhevsky, 2009; Khosla et al., 2011). We further demonstrate DTM's broad applicability with other supervisory signals and SSL frameworks and provide a deeper understanding of our design insights through ablation studies.

## 2 RELATED WORK

**Masked image modeling.** Inspired by the promising performance of masked language modeling (MLM), BEiT (Bao et al., 2022) successfully extends MLM into the computer vision domain, using an external offline tokenizer from Dall-E (Ramesh et al., 2021). iBOT (Zhou et al., 2022) jointly trains the target encoder and the online tokenizer to remove the dependency on the external tokenizer. Data2vec (Baevski et al., 2022) incorporates a momentum encoder to perform feature-level masked prediction tasks, leveraging representations from the multiple layers of neural networks. MAE (He et al., 2022) and SimMIM (Xie et al., 2022) demonstrate efficient masked image modeling by directly reconstructing masked input pixels without any tokenizer. On the other hand, several attempts have been made to exploit the pre-trained model as a tokenizer. BEiT v2 (Peng et al., 2022) pre-trains a codebook for CLIP (Radford et al., 2021) to discretize a semantic space. MVP (Wei et al., 2022b) exploits a tokenizer pre-trained with multimodal data to enhance the semantics for MIM. A line of studies (Wei et al., 2022c; Ren et al., 2023) using CLIP as a teacher to generate target representations have also been highlighted. Our method aims to utilize a teacher model more effectively, including CLIP, rather than just using it as a raw pre-trained model.

**Token aggregation methods.** Token aggregation can conceptually be categorized as a token clustering method and usually aims for efficiency. Hard clustering methods like K-Means (Lloyd, 1982), K-

Medoids (kme, 1990), and Density-Peak Clustering with K-Nearest Neighbors (DPC-KNN) (Jiang et al., 2019) enforce each data to belong to a single cluster exclusively. Bipartite matching (Karp et al., 1990) also aggregates data in a hard clustering manner, which optimizes pairs from two disjoint sets given objective function. Meanwhile, soft clustering is defined to let data belong to multiple clusters. LIT (Pan et al., 2022) employs deformable token merging layers to aggregate tokens between stages. Furthermore, some token pruning methods (Rao et al., 2021; Xu et al., 2022; Tang et al., 2022; Liang et al., 2022) can be categorized into token aggregation methods; however, they intensely focused on compressing tokens to aim for a cost-efficient vision transformer. While the above token aggregation methods have mainly been employed to boost efficiency, our approach diverges significantly. We take the concept of token aggregation to address spatially noisy target tokens in token-level supervision, thereby enhancing the efficacy of MIM in terms of precision.

## 3  PILOT STUDY

This section studies how supervisory signals from the target encoder (*e.g.*, pre-trained model) impact representation learning. In MIM or token-based SSL frameworks, these signals are typically token representations generated by the pre-trained model, which can deliver token-wise supervision. Our study is structured into two parts:

- **Defining spatial inconsistency and revealing its impact (§3.1).** We begin by visualizing predicted token representations from a pre-trained vision-language model. Fig. 1 reveals inconsistent class predictions across tokens including neighbors, which we define as *spatial inconsistency*.

  We investigate the impact of spatial inconsistency, as the token-wise noisy predictions may obscure the pre-trained model's capability. The quantitative analyses in Table 1 show that reducing the inconsistency in token representations improves classification performance without training, which highlights spatial consistency's significance.

- **Handling spatial inconsistency in representation learning (§3.2).** Our study extends the first study by exploring a pre-training scenario using a spatially inconsistent target, where a vision-language model provides token-wise supervision. We speculate that the lack of spatial coherence in the token-wise supervision hinders training, as noisy targets may weaken the supervisory signal. We then assess token aggregation methods to address the inconsistency and confirm the effectiveness of context-preserving approaches.

### 3.1  SPATIAL INCONSISTENCY

We define *spatial inconsistency* through a practical example (see Fig. 1). Given the input image in Fig. 1a, we visualize the token-wise zero-shot classification results without/with token aggregation in Fig. 1b and 1c, respectively. Correct and incorrect tokens are marked in green and red, respectively, with a gradient to darker shades of red, indicating a more significant deviation from the true class. Despite the proximity and contextual similarity among tokens, wrong tokens in the green box (bottom right) in Fig. 1b exhibit spatially inconsistent prediction results (French Bulldog, Gorilla, Bull Mastiff, Hotdog, and Newfoundland Dog) while tokens in Fig. 1c show correct or relatively consistent predictions (French Bulldog). Moreover, token aggregation for predictions improves accuracy yielding 113 correct tokens out of 196 – 31 correct tokens more than the counterpart. This highlights the spatial inconsistency among tokens from a pre-trained model, which blurs its performance and could potentially disrupt representation learning when used as a supervisory signal.

**Impact of spatial inconsistency.** To quantitatively assess spatial inconsistency's impact in a trained model, we compute ensembled token-wise predictions using global pooling, with/without token aggregations. Fundamentally, we predict class scores for each token and average these scores across

Table 1: **Addressing spatial inconsistency boosts accuracy.** We compare the ImageNet accuracy of zero-shot/linear probing via average pooled token-wise logit. Post-hoc morphed patch representations enhance accuracies, indicating that addressing spatial inconsistency improves precision. Linear probing is trained for 25 epochs, and we use the fixed half of the whole tokens for aggregation.

| Token Aggregation | Zero-shot image classification (%) | Linear probing (%) | Averaged patch-wise cosine similarity with `[CLS]` |
|---|---|---|---|
| | 26.5 | 73.2 | 0.53 |
| ✓ | 30.8 (+4.3) | 77.6 (+3.2) | 0.56 (+0.3) |

Figure 2: **Representation learning with various supervisions.** We illustrate our study's base representation learning framework along with different supervisory signal functions $f$. We evaluate four variants of the distillation target: 1) token-wise supervision (baseline); 2) downsampled supervision; 3) supervision after bipartite matching layer-wise; 4) superpixel supervision; 5) supervision by token morphing.

Table 2: **Refined supervisory signals improve representation learning.** We evaluate the ImageNet-1K linear probing accuracy of supervision types (Fig. 2). Token-wise denotes the traditional baseline, which distills token-wise supervision; only refined signals preserving context surpass it (upper rows). Lower rows showcase simpler targets, using basic methods that lose context by decreasing tokens, achieving efficiency only. We argue that the improved accuracy presumably reveals the presence of spatial inconsistency.

| Supervisory signals | Linear prob (%) | Speed↓ (ms/img) |
|---|---|---|
| (1) Token-wise (baseline) | 70.9 | 0.629 |
| (2) Superpixel clustering (Achanta et al., 2012) | 72.6 (+1.7) | 0.740 |
| (3) Token morphing (our method) | 72.2 (+1.3) | 0.653 |
| (4) Downsampling | 68.8 (-2.1) | 0.647 |
| (5) Layer-wise Bipartite Matching (Bolya et al., 2022) | 70.2 (-0.7) | 0.633 |

all tokens within the given image. When predicting with token aggregation, we group semantically relevant tokens and average their representations group-wise prior to ensembling token-wise predictions. We employ CLIP-B/16 (Radford et al., 2021) for our study and set to aggregate 98 tokens for token aggregation, half of the entire tokens.

Table 1 reports the results of zero-shot image classification, linear probing, and averaged cosine similarity with the [CLS] token on ImageNet-1K. Note that we do not perform any extra training on the model here. Our results show that using aggregated tokens consistently exceeds the performances of those without token aggregation across all metrics. This suggests aggregating tokens can address inconsistencies and lead to performance gains. The final metric we use is a continuous metric that computes patch-wise similarity; it measures the cosine similarity between the [CLS] token and each patch, which is also a patch-wise metric but continuous[1].

As observed in Table 1, the averaged similarities are computed to 0.56 vs. 0.53 for each case. This new metric, directly linked to the [CLS] token, suggests a more direct relationship between improved accuracy and reduced spatial inconsistency. The trend aligns with the other discrete metric, depicting patch-wise classification of 30.8% and 26.5% with and without token aggregation, respectively. The quantitative assessments, using both continuous and discontinuous metrics, demonstrate that token representations exhibit spatial inconsistency. Taking this further, we believe more effectively addressing inconsistency would yield a more significant impact.

### 3.2 SPATIAL INCONSISTENCY IN REPRESENTATION LEARNING

We extend the study to highlight how the spatial inconsistency in supervisory signal would affect pre-training quality. Specifically, our study is conducted in a practical representation learning scenario using a CLIP-distillation method (Peng et al., 2022) with a target encoder for token-wise distillation during training. We argue that a poor representation may suggest spatial inconsistency in the given supervisory signal. We further argue that context-preserving token aggregation (*e.g.*, smoothing tokens but preserving their amount) reduces inconsistency.

We quantitatively compare supervisory signals in token-wise distillation: 1) baseline token-wise signal; 2) superpixel clustering (Chang et al., 2023); 3) token morphing[2]; 4) downsampling; 5) layer-wise bipartite matching (Bolya et al., 2022). We pre-train ViT-B/16 for 50 epochs and linear-probe trained for 50 epochs on ImageNet-1K; CLIP-B/16 is employed for the target encoder.

---

[1] We believe a continuous metric to assess spatial inconsistency could track network responses continuously compared with the discrete classification metrics.

[2] Token morphing will be introduced in §4.1. We show its effectiveness here in advance.

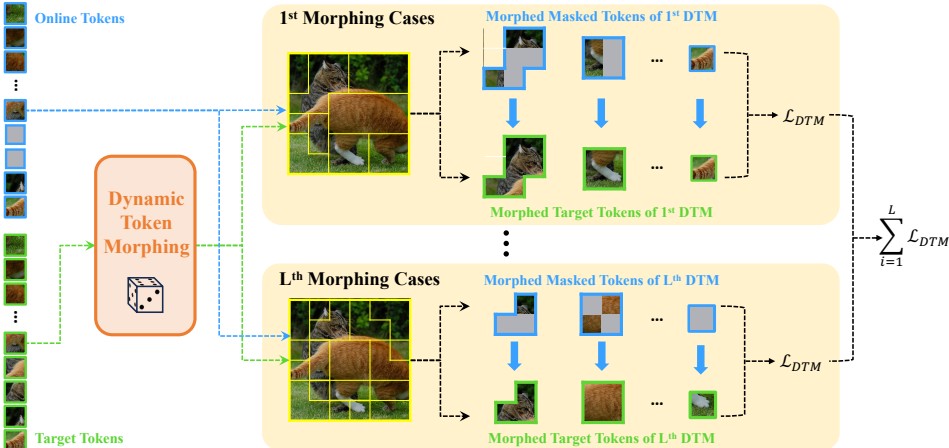

Figure 3: **Token morphing offers diverse contextualized signals.** Dynamic Token Morphing (DTM) aligns token representations by dynamically aggregating contextually related tokens to create more diverse and diversified targets. Blue and Green tokens denote the representations of the image patches processed by online and target models, respectively. Gray tokens denote masked tokens.

Table 2 exhibits distillation with refined methods (*i.e.*, superpixel clustering and token morphing) improves accuracy; this presumably confirms the presence of spatial inconsistency in the baseline token-wise supervision. However, the naive downsampling method damages intermediate representations without considering context, leading to diminished returns. While context-aware aggregation, such as the layer-wise bipartite matching, could reduce the inconsistency, it merges tokens from early layers possibly losing distinctive information without fully restoring them to their original amounts. Finally, our study provides insights for designing an improved MIM in our upcoming method, where the MIM target utilizes a vision-language model.

## 4 METHOD

We observed that 1) supervisory signals from pre-trained models often produce noisy and spatially inconsistent token-wise targets for learning, closely linked with performance degradation; 2) naive token aggregation methods could partially handle spatial inconsistency but are insufficient as a supervisory signal; 3) a well-designed method is favorable for considering context and reducing noise more effectively. Motivated by the observations, we introduce an advanced token aggregation method called *Dynamic Token Morphing* (DTM) for masked image modeling and self-supervised learning. DTM contextually aggregates tokens to derive random numbers of morphed tokens aiming to encapsulate diversified semantic information. The core idea of DTM is illustrated in Fig. 3, where the DTM module is straightforwardly added to a MIM baseline. DTM encourages a conventional token-wise MIM by aligning morphed tokens from online and target encoders by reducing spatial inconsistency while preserving context.

### 4.1 PRELIMINARY

**Token encoding.** Given an image $x$, we patchify the image into N patches $\{x_i\}_{i=1}^N$. We select positions $\mathcal{M} \subset \{1, 2, ..., N\}$ of masked patches in a block-wise manner (Bao et al., 2022; Peng et al., 2022) with a masking ratio $r \in (0, 1)$ so that $|\mathcal{M}| = \lfloor rN \rfloor$. We mask the image patches by replacing the image patches of the position in $\mathcal{M}$ to a learnable mask token $e_{[mask]}$. Specifically, the patches become $\{x_i^{\mathcal{M}}\}_{i=1}^N$, where $x_i^{\mathcal{M}} = e_{[mask]}$ for $i \in \mathcal{M}$ and $x_i^{\mathcal{M}} = x_i$ for $i \notin \mathcal{M}$. The masked patches $\{x_i^{\mathcal{M}}\}_{i=1}^N$ are concatenated with a learnable [CLS] token and fed into the online encoder $f_\theta$ with a subsequent linear head $h_\theta$ while the original patches $\{x_i\}_{i=1}^N$ are fed into the target encoder $f_\xi$, and become encoded online tokens $\{\mathbf{u}_i\}_{i=1}^N$ and encoded target tokens $\{\mathbf{v}_i\}_{i=1}^N$, respectively, where $\mathbf{u}_i = h_\theta(f_\theta(x_i^{\mathcal{M}}))$ and $\mathbf{v}_i = f_\xi(x_i)$. Here, the target encoder generates target representations for self-supervision while the online encoder learns to encode representations from the given images (Grill et al., 2020; Zhou et al., 2022; Baevski et al., 2022).

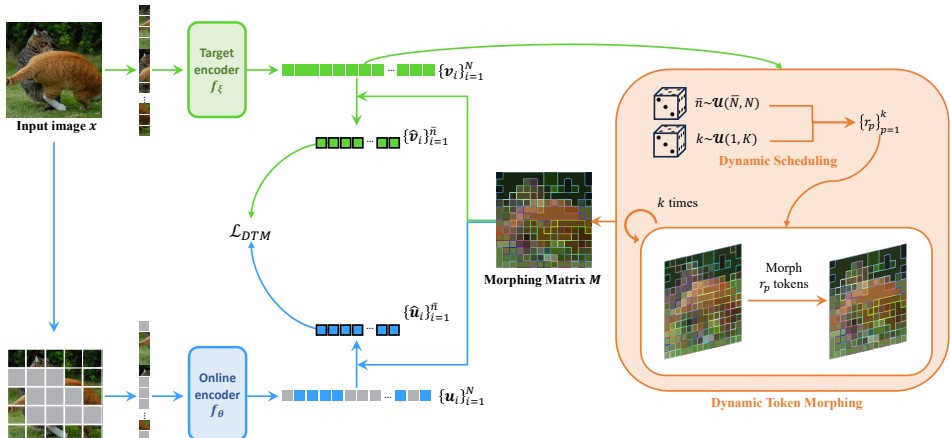

Figure 4: **Overview of Masked Image Modeling via Dynamic Token Morphing (DTM).** For a token morphing schedule of DTM, we aggregate the dynamic range of tokens using morphing matrix $M$ derived from target tokens $\{\mathbf{v}_i\}_{i=1}^{N}$. Specifically, we randomly sample a number of remaining tokens $\bar{n}$ and an iteration number $k$ to dynamically schedule token morphing (i.e., $\{r_p\}_{p=1}^{k}$), forming $\bar{n}$ morphed tokens $\{\hat{\mathbf{u}}_i\}_{i=1}^{\bar{n}}$ and $\{\hat{\mathbf{v}}_i\}_{i=1}^{\bar{n}}$. Then, we align representations of the corresponding online and target morphed tokens.

**Token morphing.** Unlike the traditional token aggregation methods, which prioritize efficiency via token reduction, token morphing preserves context to address the spatial inconsistency in token representations. It connects contextually relevant tokens to smooth them without reducing the number of tokens. We define the process with the token morphing function $\phi_R(\cdot)$ based on the morphing schedule $R$, a sequence of token numbers to morph. Here, $\phi_R$ is a general notation for a function $\phi_R : \mathbb{R}^{N \times d} \to \{0,1\}^{\bar{n} \times N}$ that calculates similarity using a matching algorithm (*e.g.*, bipartite matching[3] or K-means clustering) and returns a token morphing matrix $M = [M_{ij}] \in \{0,1\}^{\bar{n} \times N}$, where $\bar{n}$ means the number of token groups after morphing, and $d$ denotes the feature dimension. Each token is ultimately assigned a smoothed representative (*i.e.*, prototype) token (see eq. (3)).

## 4.2 MASKED IMAGE MODELING USING DYNAMIC TOKEN MORPHING

Here, we present a more advanced token morphing method, Dynamic Token Morphing (DTM), which is designed to dynamically morph tokens for MIM training. DTM's dynamic nature stems from avoiding fixed token morphing and instead simultaneously deriving multiple cases. This design is based on the insight that morphing numerous tokens enhances the denoising effect while morphing fewer tokens retains detailed token representations. Furthermore, to achieve diversified morphed tokens, the morphing process is repeated to generate multiple morphed tokens. DTM processes three key elements: 1) *dynamic scheduler* for token counts; 2) *token morphing via scheduler*; 3) *aligning morphed tokens*. The overall framework of DTM is described in Fig. 4.

**Dynamic scheduler.** DTM generates multiple morphed tokens to ensure diversity and an extensive range of token variations, as illustrated in Fig. 3. To this end, we first sample the final number of morphed tokens $\bar{n} \sim \mathcal{U}(\bar{N}, N)$ to remain after token morphing and the iteration number $k \sim \mathcal{U}(1, K)$ from uniform distributions, where $\bar{N}$ represents the minimum number of morphed tokens and $K$ denotes the maximum number of iteration for sampling. Then, we define a token count scheduler $R = \{r_p\}_{p=1}^{k}$, a sequence of token numbers $r_p \in \mathbb{N}$ that dynamically determines the number of tokens to morph for each iteration. Rather than sampling a sequence of random numbers $r_p$ that satisfies $\sum_{p=1}^{k} r_p = N - \bar{n}$, we simply divide $N - \bar{n}$ by $k$ for constant counts:

$$r_p = \begin{cases} \lfloor (N - \bar{n})/k \rfloor, & \text{if } p < k \\ N - \bar{n} - (k-1)\lfloor (N - \bar{n})/k \rfloor, & \text{if } p = k. \end{cases} \tag{1}$$

**Token morphing via dynamic scheduler.** Our token morphing function $\phi_R$ progressively morphs tokens; When the morphing target is to reduce $N$ tokens to $\bar{n}$, we design the morphing function

---

[3]Despite superpixel clustering's potential in §3.2, we tested our method employing 1) superpixel clustering and 2) context-aware bipartite matching yields accuracies of 87.1% and 87.9%, respectively (see Table E) on ImageNet-100. This prevents us from using superpixel clustering in our method.

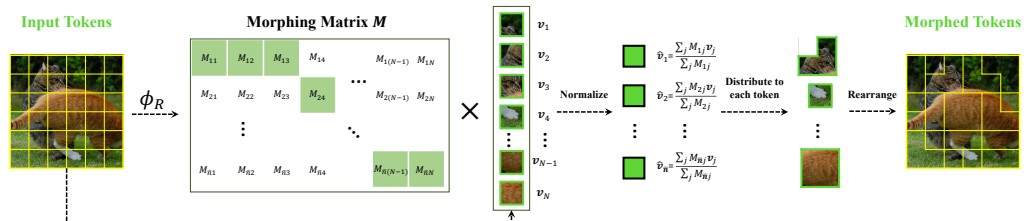

Figure 5: **Illustrative description of morphing matrix** $M = \Pi_{p=1}^{k} \bar{M}^p$. In the illustration of the morphing matrix $M$, green and white entries denote $M_{ij} = 1$ and $M_{ij} = 0$, respectively, where the $(i,j)$-th entry indicates whether the $j$-th token representations $\mathbf{v}_j$ is aggregated into the $i$-th morphed token representations $\hat{\mathbf{v}}_i$. Multiplying the morphing matrix $M$ by the token representations $\{\mathbf{v}_j\}_{j=1}^{N}$ with subsequent normalization via the number of the aggregated tokens $\sum_j M_{ij}$ yields morphed token representations $\{\hat{\mathbf{v}}_j\}_{j=1}^{N}$, as formulated in eq. (3). If we distribute the morphed tokens to their aggregated tokens and arrange the tokens, then we can achieve image representations with smoothed representations. Note that a morphing matrix is generated for each morphing case, as shown in Fig. 3.

to conduct $k$-step iterative morphing. The goal of $p$-th iteration is to reduce $r_p \in \mathbb{N}$ tokens using morphing, where $r_p$ is given to $\phi_R$ according to the dynamic scheduler $R = \{r_p\}_{p=1}^{k}$. Note that the final number of tokens is $\bar{n}$ (*i.e.*, $N - \sum_{p=1}^{k} r_p = \bar{n}$).

We eventually obtain the token morphing matrix $M$, a contextual relation among tokens, from the target token representations $\{\mathbf{v}_i\}_{i=1}^{N}$ as follows:

$$M = \phi_R\big(\{\mathbf{v}_i\}_{i=1}^{N}\big), \tag{2}$$

where $M_{ij} = 1$ indicates that the $j^{\text{th}}$ token $\mathbf{v}_j$ will be aggregated to the $i^{\text{th}}$ morphed token $\hat{\mathbf{v}}_i$, as depicted in Fig. 5. To illustrate the detailed process of $\phi_R$, suppose we have completed $(p - 1)$ number of iterations, resulting in partially morphed target token representations $\{\mathbf{v}_i^p\}_{i=1}^{N_p}$ where $N_p = N - \sum_{q=1}^{p-1} r_q$. The goal of the $p$-th iteration is to morph the $r_p$-most similar tokens. Thus, we apply the bipartite matching (Karp et al., 1990) on $\{\mathbf{v}_i^p\}_{i=1}^{N_p}$ to obtain $r_p$ number of token pairs and thereby derive the $p$-th intermediate morphing matrix $M^p = [M_{ij}^p] \in \{0,1\}^{N_{p+1} \times N_p}$, where each entry indicates whether a token is morphed or isolated. During the bipartite matching step, we split tokens into two groups, with each token in the first group matched to its closest cosine similarity counterpart in the second group. We repeat the process for $k$ iterations and gather all morphing matrices with normalization $\bar{M}^p = M^p / \sum_j M_{\cdot j}^p$ to build the morphing matrix $M = \Pi_{p=1}^{k} \bar{M}^p$, where $M \in \{0,1\}^{\bar{n} \times N}$. In addition, we let every token be assigned to a specific cluster, even in cases where it forms a single token cluster itself, and each token should retain its exclusive association with a single cluster (*i.e.*, $\sum_i \sum_j M_{ij} = \bar{n}$ and $\sum_i M_{ij} = 1$). Note that we generate the $(p+1)$-th partially morphed target token representations $\{\mathbf{v}_i^{p+1}\}_{i=1}^{N_{p+1}}$ by $\mathbf{v}_i^{p+1} = \sum_j \bar{M}_{ij}^p \mathbf{v}_j^p$ for the $(p+1)$-th iteration. The overall process of the morphing function $\phi_R$ that outputs the token morphing matrix $M$ is described in Algorithm 1 in §B through a simplified pseudo-code.

Finally, the morphed representations for both online $\{\hat{\mathbf{u}}_i\}_{i=1}^{\bar{n}}$ and target tokens $\{\hat{\mathbf{v}}_i\}_{i=1}^{\bar{n}}$ are derived based on the token morphing matrix $M$ obtained by $\phi_R$. This involves multiplying the morphing matrix $M$ with the online $[\mathbf{u}_1, \mathbf{u}_2, \ldots, \mathbf{u}_N] \in \mathbb{R}^{N \times d}$ and target token representations $[\mathbf{v}_1, \mathbf{v}_2, \ldots, \mathbf{v}_N] \in \mathbb{R}^{N \times d}$ followed by normalization with the number of aggregate tokens:

$$\hat{\mathbf{u}}_i = \frac{\sum_j M_{ij}\mathbf{u}_j}{\sum_j M_{ij}}, \quad \hat{\mathbf{v}}_i = \frac{\sum_j M_{ij}\mathbf{v}_j}{\sum_j M_{ij}}. \tag{3}$$

Note that the morphed tokens are representative tokens for each token group, with their representations being smoothed specific to their respective groups.

**Aligning morphed tokens.** We formulate the objective function by accumulating DTM losses, which aligns the representations of the corresponding online and target morphed tokens derived by DTM. The DTM loss with sampled $\bar{n}$ and $k$ is formulated as follows:

$$\mathcal{L}_{\text{DTM}}(\bar{n}, k) = \sum_{i=1}^{\bar{n}} w_i d(\hat{\mathbf{u}}_i, \hat{\mathbf{v}}_i), \tag{4}$$

Table 3: **ImageNet-1K performance comparisons**. All models were pre-trained/fine-tuned on ImageNet-1K. We evaluate the improvements in fine-tuning accuracies of competing methods using different supervisions and ours for ViT-{S/16, B/16, L/16} with a resolution of $224 \times 224$. ADE20K semantic segmentation results (Seg) using ViT-B/16 are compared as well. All our models are pre-trained for 300 epochs.

| Method | Pre-training epochs | | | Supervision | ViT-S | ViT-B | ViT-L | Seg |
|---|---|---|---|---|---|---|---|---|
| | ViT-S | ViT-B | ViT-L | | | | | |
| *Supervised models* | | | | | | | | |
| DeiT (Touvron et al., 2021) | - | - | - | Label | - | 81.8 | - | - |
| DeiT-III (Touvron et al., 2022) | - | - | - | Label | - | 83.8 | 84.2 | 49.3 |
| Cosub (Touvron et al., 2023) | - | - | - | Label | 81.5 | 84.2 | 85.3 | 49.3 |
| MaskSub (Heo et al., 2025) | - | - | - | Label | 81.7 | 84.2 | 85.3 | 50.2 |
| *Self-supervised models* | | | | | | | | |
| MoCo v3 (Chen et al., 2021) | 300 | 300 | 300 | Pixel | 81.7 | 83.2 | 84.1 | 47.3 |
| DINO (Caron et al., 2021) | 3200 | 1600 | - | Feature | 82.0 | 83.6 | - | 46.8 |
| SplistMask (El-Nouby et al., 2021) | 300 | 300 | - | Pixel+Feat. | 81.5 | 83.6 | - | 45.7 |
| BEiT (Bao et al., 2022) | 300 | 800 | 800 | DALL-E | 81.7 | 83.2 | 85.2 | 47.1 |
| iBOT (Zhou et al., 2022) | 3200 | 1600 | 1000 | Feature | 82.0 | 84.0 | 84.8 | 50.0 |
| MAE (He et al., 2022) | - | 1600 | 1600 | Pixel | - | 83.7 | 85.6 | 48.1 |
| SimMIM (Xie et al., 2022) | - | 800 | - | Pixel | - | 83.8 | - | - |
| MaskFeat (Wei et al., 2022a) | - | 1600 | 1600 | Feature | - | 84.0 | 85.7 | - |
| FD-CLIP (Wei et al., 2022c) | - | 300 | - | CLIP B/16 | - | 84.9 | - | 52.8 |
| BEiT v2 (Peng et al., 2022) | - | 300 | 300 | CLIP B/16 | - | 85.0 | 86.6 | 52.7 |
| CAN (Mishra et al., 2022) | - | 1600 | 800 | Pixel | - | 83.6 | 84.7 | - |
| data2vec (Baevski et al., 2022) | - | 800 | 1600 | Feature | - | 84.2 | 86.6 | - |
| mc-BEiT (Li et al., 2022b) | - | 800 | 800 | VQGAN | - | 84.1 | 85.6 | 47.0 |
| MVP (Wei et al., 2022b) | - | 300 | 300 | CLIP B/16 | - | 84.4 | 86.3 | 52.4 |
| SdAE (Chen et al., 2022) | - | 300 | - | Pixel | - | 84.1 | - | 48.6 |
| MSN (Assran et al., 2022) | - | 600 | - | Feature | - | 83.4 | - | - |
| BootMAE (Dong et al., 2022) | - | 800 | 800 | Pixel+Feat. | - | 84.2 | 85.9 | 49.1 |
| SemMAE (Li et al., 2022a) | - | 800 | - | Pixel | - | 83.3 | - | 46.3 |
| DeepMIM (Ren et al., 2023) | - | 300 | - | CLIP B/16 | - | 84.8 | - | - |
| AdPE (Wang et al., 2023b) | - | 1600 | - | Pixel | - | 84.4 | 86.3 | 51.5 |
| ExtreMa (Wu et al., 2023) | - | 300 | - | Feature | 81.8 | 83.7 | - | 47.9 |
| CAE (Chen et al., 2023b) | 300 | 1600 | 1600 | Pixel+Feat. | 82.0 | 83.9 | 86.3 | 50.2 |
| CMAE (Huang et al., 2023b) | - | 1600 | 1600 | Pixel+Feat. | - | 84.4 | - | 50.1 |
| ConMIM (Yi et al., 2023) | 300 | 800 | 1600 | Dictionary | 82.0 | 83.7 | 85.5 | 46.0 |
| RC-MAE (Yi et al., 2023) | 1600 | 1600 | 1600 | Pixel | 82.0 | 83.6 | 86.1 | - |
| MixedAE (Chen et al., 2023a) | - | 1600 | - | Pixel | - | 83.9 | - | 49.8 |
| SIM (Tao et al., 2023) | - | 1600 | - | Feature | - | 83.8 | - | - |
| HPM (Wang et al., 2023a) | - | 800 | 800 | Pixel | - | 84.2 | 85.8 | 48.5 |
| MIRL (Huang et al., 2023a) | - | 300 | 300 | Pixel | - | 84.1 | 85.4 | |
| CrossMAE (Fu et al., 2024) | 800 | 800 | 800 | Pixel | 79.3 | 83.7 | 85.4 | - |
| dBOT (Liu et al., 2024) | - | 1600 | 1600 | Feature | - | 84.5 | 86.6 | 49.5 |
| LUT (Kim et al., 2024) | 400 | 1600 | 1600 | Pixel | 82.0 | 84.2 | 86.0 | 49.5 |
| MI-MAE (Huang et al., 2025) | - | 400 | - | Pixel | - | 84.1 | - | 49.3 |
| DTM (ours) | - | 300 | 300 | 300 CLIP B/16 | **83.2** | **85.4** | **86.7** | **53.1** |

where $d(\cdot)$ is a distance function and $w_i = \sum_j M_{ij}$ is a number of tokens aggregated for the $i^{\text{th}}$ online and target morphed tokens. Here, we utilize $w_i$ to consider all tokens aggregated for the morphed tokens. The DTM loss can be extended to token-wise or image-level losses when $\bar{n} = N$ or $\bar{n} = 1$, respectively. We adopt Cosine distance for the distance function in eq. (4) for all the DTM losses. To further enhance the dynamic nature of our method, we apply multiple DTM losses, each derived from its corresponding morphing case. The final objective function is the summation of all DTM losses, which is defined as:

$$\min_\theta \sum_{l=1}^{L} \mathcal{L}_{\text{DTM}}(\bar{n}_l, k_l) \quad \text{s.t. } \bar{n}_l \sim \mathcal{U}(\bar{N}_l, N) \text{ and } k_l \sim \mathcal{U}(1, K_l), \tag{5}$$

where $L$ denotes the total number of simultaneously employed DTM losses.

## 5 EXPERIMENT

This section first reports the ImageNet-1K classification and ADE20K segmentation performances. We explore DTM's strengths in terms of its efficiency and applicability. Finally, we study the impact of DTM's dynamic nature and pre-training capability. Note that additional results on DTM pre-trained models' transfer learning (§C), further DTM's applicability (§D), more empirical studies (§E), and the implementation details (§F) are in Appendix.

Table 4: **Efficiency of context-aware token aggregations**. We report fine-tuning accuracies and throughputs for each configuration, which are pre-trained with ViT-B/16. We compare DTM with Bipartite matching, DTM with K-means clustering, and layer-wise K-means clustering. For the layer-wise K-means clustering, we use constant numbers of clusters and iterations. DTMs both outperform the baseline with large margins. Moreover, DTM with K-means clustering surpasses layer-wise K-means clustering, demonstrating the superiority of DTM.

| Case | Throughput (image/s) | FT (%) |
|---|---|---|
| Baseline | **1458** | 84.3 |
| Layer-wise K-means clustering | 1265 | 85.1 |
| DTM (K-means clustering) | 489 | **85.4** |
| DTM (Bipartite matching) | 1446 | **85.4** |

## 5.1 IMAGENET-1K CLASSIFICATION

We compare the fine-tuning accuracy of our method with previous state-of-the-art self-supervised methods on ImageNet-1K (Russakovsky et al., 2015). The comparisons include supervised learning and SSL methods with various supervisory signals. When the target model is CLIP, we only compare models pre-trained with CLIP B/16 for 300 epochs for a fair comparison. Table 3 reports the fine-tuning accuracies of ViT-S/B/L backbones. Our baseline simply employs negative cosine loss with a vanilla CLIP model as the target model. We observe that our MIM pre-trained by DTM achieves 83.2%, 85.4%, and 86.7% top-1 accuracies with ViT-S/16, ViT-B/16, and ViT-L/16, respectively, which outperforms state-of-the-art performances across the scales. Specifically, our method surpasses MVP (Wei et al., 2022b), DeepMIM (Ren et al., 2023), and BEiT v2 (Peng et al., 2022) by 1.0%p, 0.6%p, and 0.4%p on ViT-B/16, respectively. Moreover, our method outperforms other methods that leverage diverse supervision, demonstrating our method's superiority among self-supervised learning methods.

Additionally, we extend DTM's pre-training to 800 epochs, which improves fine-tuning accuracy of **85.5%** on ImageNet-1K. This result highlights 1) DTM merits longer pre-trainings and 2) DTM trained for 800 epochs surpasses others trained for 1600 epochs such as BEiT v2 (Peng et al., 2022). We believe even longer pre-training of DTM would lead to further improvements.

## 5.2 ADE20K SEMANTIC SEGMENTATION

We further evaluate semantic segmentation on ADE20K (Zhou et al., 2017) to verify the transferability of our pre-trained model. We follow the training and evaluation protocol (He et al., 2022); a model is fine-tuned for 160K iterations using UperNet (Xiao et al., 2018) with a batch size of 16 and a resolution of $512 \times 512$. We initialize UperNet with our pre-trained ViT-B/16. Detailed hyperparameters for semantic segmentation fine-tuning can be found in Appendix. The first right column in Table 3 shows the mIoU performance comparison. Our method also outperforms the previous state-of-the-art results with a margin of 0.3%p, validating its superiority over other SSL methods. This result signifies that our method effectively enhances discriminability for dense prediction tasks.

## 5.3 EFFECTIVENESS OF OUR METHOD

**Efficiency.** Table 4 shows that bipartite matching is efficient and effective, significantly boosting the accuracy (+1.1%p) with only a 1% speed loss. While K-means (Lloyd, 1982) also exhibits considerable improvements, it significantly degrades the training speed. Layer-wise K-means shows accelerate the K-means method by aggregating tokens within layers at the cost of degraded representations, leading to lower accuracy.

**Transferability.** We verify the improved transferability of our pre-trained model. We compare fine-tuning accuracies of the baseline and our proposed model on iNaturalist datasets (Van Horn et al., 2018), which are highly imbalanced with different numbers of images per class, and Fine-Grained Visual Classification (FGVC) datasets. Table C and Table D in the Appendix show our DTM loss significantly improves the baseline with large margins, which reveals enhanced transferability.

## 5.4 EMPIRICAL ANALYSIS

**Impact of the dynamic nature of DTM.** Table 5 presents an ablation study on the effectiveness of the dynamic nature of DTM. We employ ViT-B/16 (Dosovitskiy et al., 2021) with a resolution of

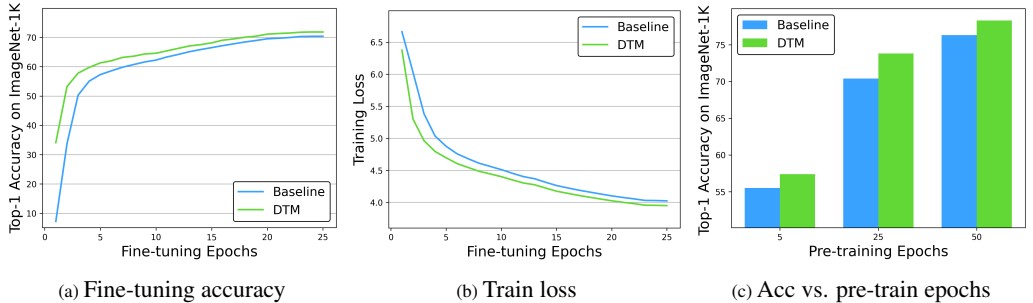

(a) Fine-tuning accuracy  (b) Train loss  (c) Acc vs. pre-train epochs

Figure 6: **Visualizations with DTM.** We plot (a) top-1 accuracies and (b) training losses during fine-tuning on ImageNet-1K for models pre-trained by Dynamic Token Morphing (DTM) versus its baseline. (c) confirms the impact of pre-training epochs for DTM over the baseline. We train the ViT-B/16 architectures with a resolution of $224 \times 224$. In both (a) and (b), DTM consistently exhibits a substantial gap compared to the baseline during the entire fine-tuning phase, indicating that DTM offers stronger supervision that facilitates training. DTM consistently improves the baseline regardless of the pre-training epochs, as shown in (c).

Table 5: **Empirical study on DTM's dynamic nature.** We investigate the efficacy of dynamic scheduling in DTM, which reveals its significant contribution to our method.

| Method | $\mathcal{L}_{\text{Token\_Morphing}}$ | Dynamic | FT Acc. (%) |
|---|---|---|---|
| Baseline | - | - | 84.3 |
| DTM | ✓ | - | 84.0 (-0.3) |
| DTM | ✓ | ✓ | **85.4** (+1.1) |

$224 \times 224$. The models are pre-trained for 300 epochs and fine-tuned for 100 epochs on ImageNet-1K (Russakovsky et al., 2015). For token morphing without dynamic scheduling, half of the total 196 tokens are aggregated for each image. Table 5 exhibits that token morphing with the dynamic scheduler significantly improves the baseline while its absence incurs performance degradation, which highlights the importance of dynamic nature of DTM.

**Improved training trends with DTM pre-training.** We analyze the effectiveness of DTM pre-training and the baseline pre-training method using the MIM's token-wise objective in Fig. 6. All the methods employ CLIP representations for the MIM targets. As shown in Fig. 6a, the fine-tuning accuracies of DTM surpass the baseline and starts from a significantly higher initial point. Furthermore, Fig. 6b shows that the model pre-trained by DTM exhibits a lower fine-tuning loss than the baseline model suggesting presumably better loss convergence. Finally, Fig. 6c reveals the consistent advantages of DTM pre-training across various pre-training epochs.

## 6  CONCLUSION

We have introduced a novel masked image modeling method based on the proposed token morphing to address spatially inconsistent target representations during pre-training. We have first revealed the existence and impacts of spatial inconsistency in target representations. Specifically, we have qualitatively observed spatial inconsistency among tokens from pre-trained models despite proximity and contextual similarity. We have then investigated a representation learning scenario through accuracy metrics like zero-shot or linear classification. Our study has validated that context-preserving token aggregation methods enhance the pre-training capability of the target, while arbitrary aggregation, like downsampling, disrupts it. Based on the observations, we have proposed Dynamic Token Morphing (DTM), which dynamically aggregates contextually associated tokens with randomness and iteratively generates diverse sets of morphed tokens. MIM training is performed by aligning the representations of morphed tokens from the online and target encoders. Our extensive experiments have verified its scalability and performance superiority. We have further validated its applicability and effectiveness through empirical studies. We believe that our insights on token aggregation considering context preservation could boost supervisory signals in representation learning and beyond, with the potential for broader future applications and research directions.

**Limitation.** Despite the potential of DTM, we have verified its applicability only up to ViT-L/16. Resource limitations restricted larger-scale experiments such as with ViT-G.

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

# Appendix

This appendix includes additional experimental analyses of our proposed method:

- §A: Additional examples of spatial inconsistency among patches from pre-trained models
- §B: An algorithm for Token Morphing Function
- §C: Transferability of DTM on iNaturalist and Fine-Grained Visual Classification (FGVC) datasets
- §D: Applicability of DTM on other targets and SSL frameworks
- §E: Ablation studies on compatibility with superpixel algorithms (Achanta et al., 2012) into DTM, the number of morphing schedules, effects of randomness in the number of morphing tokens, randomness in gradual token morphing, and target normalization
- §F: Implementation details for both pre-training and fine-tuning on ImageNet-1K (Russakovsky et al., 2015) and fine-tuning on ADE20K (Zhou et al., 2017)

## A    MORE EXAMPLES ON SPATIAL INCONSISTENCY

**EVA-CLIP.**    We extend our analysis to explore the spatial inconsistency of visual token predictions produced by other supervisory models. We employ a strong and larger-scale model: EVA-01-CLIP-g/14 (Sun et al., 2023), which is the teacher model for EVA-02 (Fang et al., 2024). Following the analysis in Fig. 1, we visualize token-wise zero-shot classification results with and without token aggregation. Consistent with our earlier approach, we aggregate 128 tokens, corresponding to half of the total tokens. Fig. A demonstrates that token-wise zero-shot predictions without token aggregation exhibit spatially inconsistent token-wise predictions compared to those with token aggregation, similar to the behavior observed in the CLIP case. Zero-shot accuracies on ImageNetV2 (Recht et al., 2019) computed by the token-wise ensemble prediction via global pooling, follow a similar trend, where aggregation enhances zero-shot performance (53.1% vs. 51.2%) reflecting the capability of supervisory signals. This suggests that even a stronger pre-trained model can benefit from token aggregation when used as a supervisory signal.

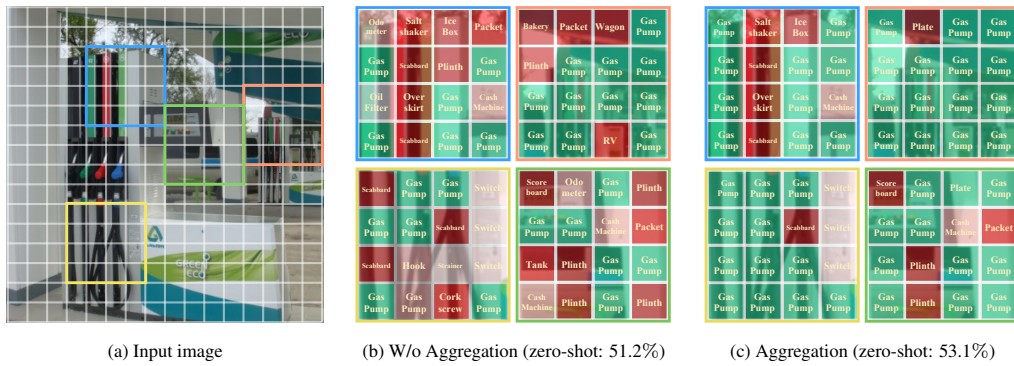

(a) Input image          (b) W/o Aggregation (zero-shot: 51.2%)          (c) Aggregation (zero-shot: 53.1%)

Figure A: **Visualization of spatial inconsistency using EVA-CLIP.** We present token-wise prediction results of a sample for zero-shot image classification using EVA-01-CLIP-g/14 (Sun et al., 2023) to visualize the spatial inconsistent predictions among patches. For the given input images, we visualize the predicted classes for each patch in the four bounding boxes through token-wise zero-shot classification with and without token aggregation, respectively. Precisely, we depict the differences between the predicted classes and the ground-truth class by varying the lightness of red colors. We observe that the prediction results of the morphed tokens are more likely to align with the input patches, leading to a reduction in the spatial inconsistency incurred by token-wise predictions.

**CLIP.**    We further investigate the spatial inconsistency of patch representations generated by a pre-trained model across various samples. We visualize the token-wise zero-shot classification results

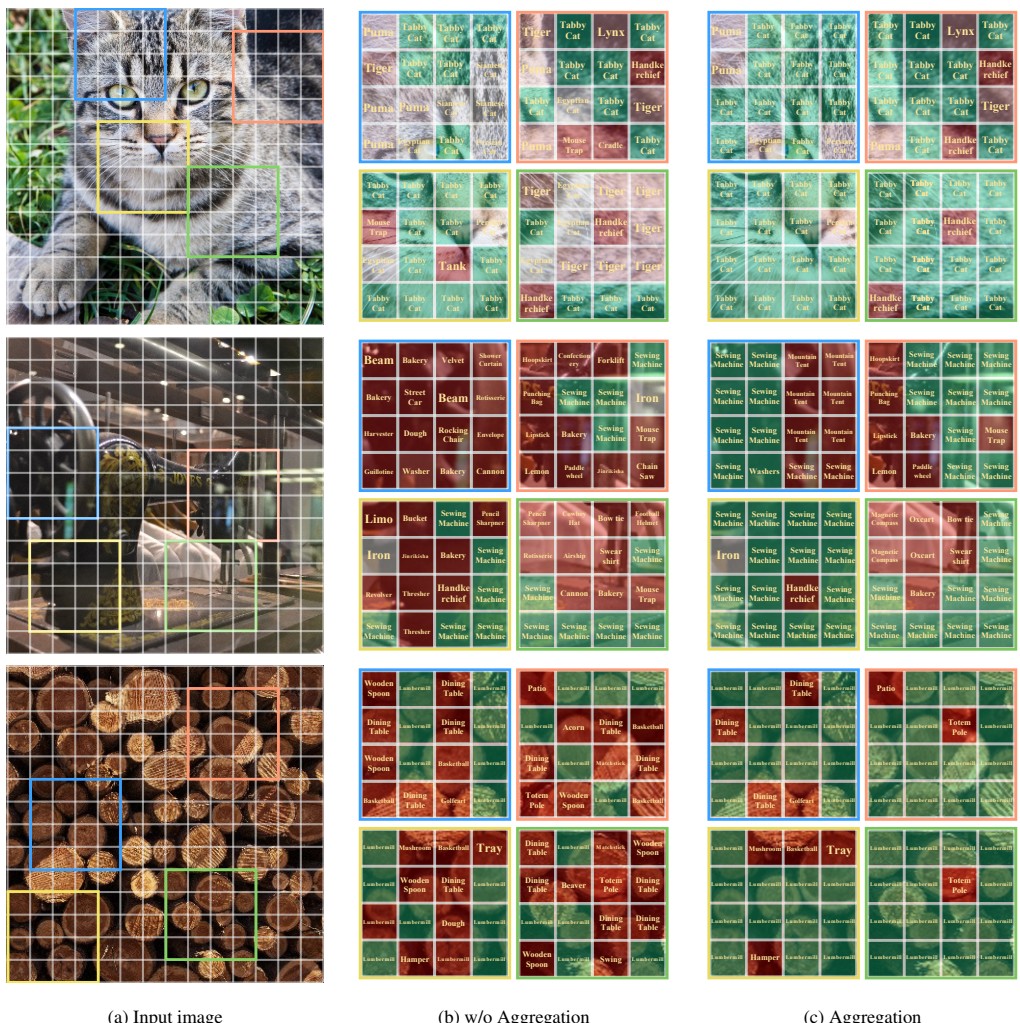

(a) Input image          (b) w/o Aggregation          (c) Aggregation

Figure B: **Visualization of spatial inconsistency using CLIP.** We present patch-wise prediction results of a sample for zero-shot image classification to visualize the spatial inconsistent predictions among patches. For the given input images in (a), (b), and (c), we visualize the predicted classes for each patch in the bounding boxes through patch-wise zero-shot classification with and without token aggregation, respectively. Precisely, we depict the differences between the predicted classes and the ground-truth class by varying the lightness of red colors. We observe that the prediction results of the morphed tokens are more likely to align with the input patches, leading to a reduction in the spatial inconsistency incurred by patch-wise predictions.

without and with token aggregation using CLIP (Radford et al., 2021). Specifically, correct and incorrect tokens are marked in green and red, respectively, with a gradient to darker shades of red, indicating a more significant deviation from the true class. As shown in Fig. B, patch representations without token aggregation reveal spatially inconsistent token-wise predictions compared to the prediction results with token aggregation, which reveals the spatially inconsistent representations among patches. In addition, predictions with token aggregation exhibit significantly more correctly predicted patches than predictions without token aggregation.

## B  ALGORITHM FOR TOKEN MORPHING FUNCTION

In this section, we describe the generation process of the morphing matrix $M$ for the token morphing function $\phi_R$. This function utilizes from the target representations $\{v_i\}_{i=1}^N$, the dynamic scheduler $R$, and the iteration number $k$ in Algorithm 1.

---

**Algorithm 1:** Token Morphing Function ($\phi_R$)

---

1: **input:** token representation $\{\mathbf{v}_i\}_{i=1}^N$, iteration $k$, scheduler $R = \{r_p\}_{p=1}^k$
2: **define** $n \leftarrow N$
3: **define** $\mathbf{v}_i^0 \leftarrow \mathbf{v}_i$ for $i \in [1, N]$
4: **for** $p \in \{1, \ldots, k\}$ **do**                                        # k-iterative morphing
5:     $M^p \leftarrow \text{BIPARTITEMATCHING}(\mathbf{v}^p, n)$
6:     $\bar{M}_{ij}^p \leftarrow M_{ij}^p / \sum_{j'=1}^n M_{ij'}^p$ for all $i, j$                # Normalize
7:     $\mathbf{v}_i^{p+1} \leftarrow \sum_{j=1}^n \bar{M}_{ij}^p \mathbf{v}_j^p$ for $i \in [1, n - r_p]$        # Morph matched tokens
8:     $n \leftarrow n - r_p$
9: **return** $M = \Pi_{p=1}^k \bar{M}^p$

10: **function** BIPARTITEMATCHING($\mathbf{v}^p, n$)                   # Standard bipartite matching algorithm
11:     $(\mathcal{S}_1^p, \mathcal{S}_2^p) \leftarrow \text{random\_split}([1, 2, \ldots, n])$     # Split for Bipartite matching
12:     $\text{sim} \leftarrow [\text{Sim}(\mathbf{v}_i^p, \mathbf{v}_j^p) \text{ for } (i, j) \in \mathcal{S}_1^p \times \mathcal{S}_2^p]$     # Measure similarity
13:     $\sigma \leftarrow \text{sort}(\text{sim}, \text{order}=\text{'descending'})[r_p]$      # Threshold for top-$r_p$ similarity
14:     $M_{ij}^p \leftarrow 1; M^p \leftarrow M^p \backslash M_{j.}^p \text{ s.t. } \text{Sim}(\mathbf{v}_i^p, \mathbf{v}_j^p) \geq \sigma, (i, j) \text{ in } \mathcal{S}_1^p \times \mathcal{S}_2^p$
15:     **return** $M^p$
16: **end function**

---

Table A: **Applicability of DTM on SLIP.** We apply DTM to SLIP (Mu et al., 2021), a more improved language-image pre-trained model to demonstrate our method's applicability beyond CLIP. We employ ViT-{S/16, B/16, L/16} with a resolution of $224 \times 224$. All models are pre-trained for 300 epochs on ImageNet-1K.

| Target models | Method | ViT-S | ViT-B | ViT-L |
|---|---|---|---|---|
| SLIP | Baseline | 81.8 | 84.0 | 85.7 |
| | DTM | **82.1** (+0.3) | **84.5** (+0.5) | **86.1** (+0.4) |

## C    ASSESSING TRANSFER LEARNING

**iNaturalist datasets.** We verify the improved transferability of our pre-trained model. We compare fine-tuning accuracies of the baseline and our proposed model on iNaturalist 2018, iNaturalist 2019, and mini iNaturalist 2021 (Van Horn et al., 2018), which are highly imbalanced with different numbers of images per class. All the models are ViT-B/16 with a resolution of $224 \times 224$. Following the protocol (Kornblith et al., 2019), we perform grid searches on learning rates and weight decay and report the maximum accuracy and the mean and standard deviation of the accuracies. Table C shows our DTM loss significantly improves the baseline by large margins, demonstrating enhanced transferability.

**Fine-Grained Visual Classification (FGVC) datasets.** We further validate the transferability of our method. Following the evaluation protocol as above, we conduct comparisons on FGVC datasets. Specifically, we evaluate fine-tuning accuracies on Birds, CUB-200, CIFAR-10, CIFAR-100, and Dogs through grid searches with different learning rates and weight decays. As shown in Table D, our method outperforms the baseline overall across the datasets, which shows superior transferability and tuning robustness.

## D    MORE INVESTIGATION ON EFFECTIVENESS OF OUR METHOD

**On other targets.** To confirm our method's applicability, we apply DTM with the SLIP (Mu et al., 2021) to verify its effectiveness across different target models. We compare fine-tuning accuracies of the baseline and our method pre-trained by target patches from SLIP. Specifically, we generate target tokens using SLIP along with its projector. We pre-train ViT-B/16 with a resolution of $224 \times 224$ for 300 epochs and fine-tune for 100 epochs on ImageNet-1K (Russakovsky et al., 2015). As shown in Table A, our DTM successfully improves the fine-tuning accuracy of the baselines pre-trained with SLIP by 0.5%p, which reveals its general applicability beyond CLIP (Radford et al., 2021).

**Applicability to SSL frameworks.** We apply DTM to BEiT v2, MAE with the CLIP teacher (Radford et al., 2021), and BYOL (Grill et al., 2020) to verify its applicability to various SSL frameworks. As shown in Table B, our DTM successfully improves the fine-tuning accuracy of MAE +

Table B: **Applicability of our DTM on various SSL frameworks.** We train DTM with other SSL methods to show its broader applicability. We adopt MAE (He et al., 2022) with the CLIP teacher (Radford et al., 2021), BEiT v2 (Peng et al., 2022), and BYOL (Grill et al., 2020) to cover general SSL frameworks. We employ the ViT-B/16 architecture with a resolution of $224 \times 224$. All the models are pre-trained for 100 epochs.

| Framework | Method | FT (%) |
|---|---|---|
| Feature MIM (CLIP teacher) | MAE + CLIP | 82.6 |
| | MAE + CLIP + DTM | **83.2** (+0.6) |
| | BEiT v2 | 84.2 |
| | BEiT v2 + DTM | **84.4** (+0.2) |
| Image-level SSL | BYOL | 81.7 |
| | BYOL + DTM | **82.1** (+0.4) |

Table C: **Transfer learning results on iNaturalists**. We further present the end-to-end fine-tuning accuracies on the iNaturalist 2018, iNaturalist 2019, and mini iNaturalist 2021 datasets (Van Horn et al., 2018). We report the best results along with the mean $\pm$ std of the set of accuracies obtained from grid searches for each method.

| Method | iNat 2018 | iNat 2019 | iNat 2021-mini |
|---|---|---|---|
| Baseline | 75.0 (74.6±0.6) | 81.1 (79.8±1.0) | 75.8 (75.2±0.6) |
| DTM (Ours) | **78.5** (77.4±0.6) | **81.9** (81.2±0.6) | **78.4** (77.5±0.6) |

Table D: **Transfer learning results on Fine-Grained Visual Classification (FGVC) datasets.** We present the end-to-end fine-tuning accuracies on multiple datasets, reporting the best results along with the mean $\pm$ std of the accuracies from grid searches. Our Dynamic Token Morphing (DTM) outperforms the baseline at the best accuracies overall.

| Method | Birds | CUB-200 | CIFAR-10 | CIFAR-100 | Dogs | Average |
|---|---|---|---|---|---|---|
| Baseline | 87.3 (86.5±0.6) | 87.1 (86.8±0.6) | 99.2 (99.1±0.0) | 92.0 (91.9±0.3) | 86.9 (86.8±0.1) | 90.5 |
| DTM (Ours) | 88.8 (88.2±0.4) | 88.8 (88.1±0.4) | 99.3 (99.2±0.0) | 92.3 (92.1±0.2) | 87.9 (87.8±0.2) | 91.4 (+0.9) |

CLIP, BEiT v2, and BYOL by 0.5%p, 0.2%p, and 0.4%p, respectively, which reveals its general applicability to various frameworks beyond our baseline.

# E  ADDITIONAL EMPIRICAL STUDIES

## E.1  COMPATIBILITY WITH SUPERPIXEL ALGORITHMS

We perform additional experiments to use the concept of superpixels (Achanta et al., 2012; Chang et al., 2023) in our method, both directly to tokens (layer-wise superpixel) and can be combined with DTM. The layer-wise superpixel method uses constant numbers of superpixels and iterations. Table E shows superpixel-based methods enjoy notable gains but do not exceed the bipartite matching one. While the layer-wise superpixel method also utilizes superpixels, the layer-wise token aggregation across the encoder layers in the bipartite matching approach risks harming intermediate representations during encoding

## E.2  ABLATION STUDIES ON RANDOMNESS IN DTM

**The number of dynamic schedules.** Our method can further enhance the diversity of target morphed tokens by employing multiple schedules within a single iteration. Thus, we study the effects of learning diverse morphed tokens derived from multiple morphing schedules simultaneously. We compare the fine-tuning accuracy of models pre-trained by our DTM, varying the number of schedules. We pre-train ViT-B/16 with a resolution of $224 \times 224$ for 100 epochs and fine-tune for 100 epochs on ImageNet-1K (Russakovsky et al., 2015) and ImageNet-100 (Russakovsky et al., 2015). Table F reveals that exploring diverse target morphed tokens improves the representation capability of pre-trained models, leading to increased fine-tuning accuracies of at least 0.1%p and 0.2%p on ImageNet-1K and ImageNet-100 compared to the model pre-trained by the single DTM approach, respectively. However, the goal of experiencing diverse morphed tokens at once appears to be attained with double scheduling, resulting in no additional gains in performance through further

Table E: **Applicability of DTM on the superpixel algorithm.** All the studies report fine-tuning accuracies for each configuration pre-trained using ViT-B/16. All models are pre-trained for 100 epochs on ImageNet-100. Here, the Layer-wise superpixel method denotes a token reduction approach that generates superpixel tokens across layers. While the superpixel algorithm is applicable to DTM, bipartite matching exhibits the best performance, demonstrating the superiority of our design choice. The default settings for the study are marked in gray .

| Method | Fine-tuning (%) |
|---|---|
| Baseline | 79.5 |
| Layer-wise superpixel (Achanta et al., 2012) | 86.7 |
| DTM (Superpixel clustering) (Chang et al., 2023) | 87.1 |
| DTM (Bipartite matching) | **87.9** |

Table F: **Ablation study on the number of DTM schedules**. We study the effect of exploring diverse morphed tokens through multiple scheduling. We report fine-tuning accuracies for each configuration, which are pre-trained with ViT-B/16. All the backbones are pre-trained for 100 epochs on ImageNet-1K Russakovsky et al. (2015). Simultaneous exploration using multiple morphing schedules further enhances the performance. The default settings for the study are marked in gray .

| Method | Case | IN-1K Fine-tuning (%) | IN-100 Fine-tuning (%) |
|---|---|---|---|
| Baseline | | 83.5 | 79.5 |
| DTM | 1 | 84.8 | 87.6 |
| | 2 | **84.9** | **87.9** |
| | 3 | 84.8 | 87.8 |
| | 4 | 84.8 | 87.8 |

exploration. Given that utilizing two schedules yields the best and is most efficient among all other multiple scheduling options, we adopt the double morphing scheduling approach.

**Randomness in the number of morphing tokens.** Our DTM randomizes the number of morphing tokens since fixing this number does not adequately generate diverse morphed tokens. To verify the effectiveness of the randomness, we compare the fine-tuning accuracies of models pre-trained using DTM with random or fixed numbers of morphing tokens. We pre-train and fine-tune the models for 100 epochs on ImageNet-100 (Russakovsky et al., 2015). As reported in the 2nd and 5th rows of Table G, exploring diverse morphed tokens improves the fine-tuning accuracy by 0.5%p. This result demonstrates the impact of varying the number of morphing tokens.

**Randomness in gradual token morphing.** As morphed tokens can vary by the number of morphing iterations, we apply randomness to token morphing iterations. We compare fine-tuning accuracies of the models pre-trained with random or fixed iteration numbers for morphing to validate the effect of randomness. As shown in the 3rd and 5th rows of Table G, randomly selecting the iteration number for morphing enhances the performance, confirming its effectiveness.

### E.3    ON TARGET NORMALIZATION

Target normalization is proven to have a significant impact on MAE (He et al., 2022). Thus, we verify the impact of target normalization on our DTM. We employ ViT-B/16 with a resolution of $224 \times 224$ for comparison. As shown in the 4th and 5th rows of Table G, target normalization does not yield a positive effect on our DTM, resulting in a decrease in fine-tuning accuracy from 87.9% to 87.7%.

### E.4    FURTHER STUDIES

We conduct ablation studies on loss functions and the range of token morphing steps. We also examine the impacts of the number of morphing schedules. We pre-train ViT-B/16 with a resolution of $224 \times 224$ for 100 epochs and fine-tune for 100 epochs on ImageNet-1K (Russakovsky et al., 2015).

Table G: **Ablation study on various configurations.** We investigate the effectiveness of our design choices for DTM: morphing a random number of tokens, gradual morphing by multiple morphing steps, and target normalization. We report fine-tuning accuracies for each configuration. We adopt ViT-B/16 with a resolution of $224 \times 224$. All the models are pre-trained for 100 epochs on ImageNet-100 Russakovsky et al. (2015). While other configurations improve the baseline well, our design yields the best accuracy. We mark the default settings for the study in  gray .

| Method | Configurations | Fine-tuning (%) |
|---|---|---|
| Baseline | | 79.5 |
| DTM (ours) | Fixed number of morphing tokens | 87.4 |
| | Single step morphing | 87.7 |
| | + Target normalization He et al. (2022) | 87.7 |
| | Default | **87.9** |

Table H: **Ablation studies on a single DTM schedule**. We perform ablation studies on loss functions, ranges of token morphing steps, and target normalization. All the studies report fine-tuning accuracies for each configuration pre-trained using ViT-B/16. All models are pre-trained for 100 epochs. The default settings for the study are marked in  gray .

(a) **Loss function**

| Case | Fine-tuning (%) |
|---|---|
| $\ell_1$ | 84.5 |
| $\ell_2$ | 84.6 |
| Smoothed $\ell_1$ | 84.4 |
| Cosine distance (Cos) | **84.8** |

(b) **Morphing steps**

| Case | Fine-tuning (%) |
|---|---|
| $\mathcal{U}(1, 7)$ | 84.7 |
| $\mathcal{U}(1, 14)$ | **84.8** |
| $\mathcal{U}(1, 28)$ | 84.7 |

**Loss function.** We compare various options for the loss function in our method. We compared $\ell_1$, $\ell_2$, smoothed $\ell_1$, and cosine distance. As shown in Table Ha, the model pre-trained using cosine distance outperforms the models with other distance functions.

**Range of token morphing steps.** We compare fine-tuning accuracies of the pre-trained models while varying the ranges that randomly sample the number of morphing iterations in Table Hb. While our DTM works for all the sampling ranges, K=14 works best.

## F  IMPLEMENTATION DETAILS

**Pre-training on ImageNet-1K.** The pre-training recipe for DTM mainly follows the recipe of BEiT v2 (Peng et al., 2022). Table I reports the implementation details for pre-training. We train our framework with ViT-S/16, ViT-B/16, and ViT-L/16 for 300 epochs using AdamW with momentum (0.9, 0.98) and a batch size of 1024. We use a learning rate of $1.5 \times 10^{-4}$ with cosine decay and warmup 10 epochs. We employ the CLIP base models (Radford et al., 2021) with its visual projector as a target model across all scales of ViT. Block-wise masking is used with a ratio of 0.4 following (Bao et al., 2022; Peng et al., 2022). Cosine distance is used as a distance metric for the objective according to an ablation study in Appendix. We adopt the hyperparameters for ViT-B/16 and ViT-L/16 pre-training from BEiT v2. Specifically, we use layer scales of 0.1 and $1 \times 10^{-5}$ for ViT-B/16 and ViT-L/16, respectively. We employ both relative positional embeddings and shared relative positional embeddings. The maximum gradient value is constrained to 3.0. We apply color jittering followed by random resizing and cropping for data augmentation. The hyperparameters for ViT-S/16 replicate the settings of ViT-B/16. We also pre-train various SSL frameworks through our DTM with the same fundamental setups.

**Fine-tuning on ImageNet-1K.** We fine-tune our pre-trained models on ImageNet-1K (Russakovsky et al., 2015) by default following the standard protocol (He et al., 2022; Peng et al., 2022). Specifically, pre-trained ViT-S/-B/-L are fine-tuned for 300, 100, and 50 epochs, respectively. Optimization is performed with AdamW using a weight decay of 0.05. We use a layer-wise learning rate decay of 0.6 for ViT-S and ViT-B and 0.8 for ViT-L. Learning rate is set to $5 \times 10^{-4}$ with a linear warmup for 10 epochs for ViT-S and ViT-B and 5 epochs for ViT-L. We adopt commonly used values for RandAugment, Mixup, Cutmix, and Label Smoothing. On the other hand, we employ relative positional

embeddings. Stochastic depth is applied with values of 0.1, 0.1, and 0.2 for ViT-S/16, ViT-B/16, and ViT-L/16, respectively. The overall recipe is detailed in Table J.

**Fine-tuning on ADE20K.** Table K summarizes the fine-tuning recipe of ViT/16 for the semantic segmentation task on ADE20K (Zhou et al., 2017). We employ AdamW with momentum (0.9, 0.999) and warm-up for 1500 iterations. The learning rate is linearly scheduled with a value of $5 \times 10^{-5}$ We apply layer-wise learning rate decay of 0.75, stochastic depth of 0.1, and weight decay of 0.05. The model is fine-tuned using 8 V100-32GB GPUs.

**Transfer learning.** We follow the fine-tuning recipes for DTM to conduct transfer learning to iNaturalist datasets, including iNaturalist 2018, iNaturalist 2019, and mini iNaturalist 2021 (Van Horn et al., 2018) and FGVC datasets, including Birds, CUB-200, CIFAR-10, CIFAR-100, and Dogs. However, we additionally perform grid searches of learning rates and weight decay. Specifically, we fine-tune the models with learning rates of $2.5 \times 10^{-5}$, $5 \times 10^{-5}$, and $1 \times 10^{-4}$ and weight decays of 0.05 and 0.1.

**Applicability on various SSL frameworks.** When pre-training and fine-tuning models with MAE (He et al., 2022), BEiT v2 (Peng et al., 2022), and BYOL (Grill et al., 2020), we follow their vanilla training recipes. We pre-train and fine-tune ViT-B/16 for 100 epochs. However, we use CLIP (Radford et al., 2021) target features instead of patchified images to pre-train MAE.

Table I: Hyperparameters for pre-training on ImageNet-1K.

| Hyperparameters | ViT-S/16 | ViT-B/16 | ViT-L/16 |
|---|---|---|---|
| Layers | 12 | 12 | 24 |
| Hidden size | 384 | 768 | 1024 |
| FFN inner hidden size | 1536 | 3072 | 4096 |
| Attention heads | 6 | 12 | 16 |
| Layer scale | 0.1 | 0.1 | 1e-5 |
| Patch size | | $16 \times 16$ | |
| Relative positional embeddings | | ✓ | |
| Shared relative positional embeddings | | ✓ | |
| Training epochs | | 300 | |
| Batch size | | 1024 | |
| Adam $\beta$ | | (0.9, 0.98) | |
| Base learning rate | | 1.5e-4 | |
| Learning rate schedule | | Cosine | |
| Warmup epochs | | 10 | |
| Gradient clipping | | 3.0 | |
| Dropout | | ✗ | |
| Drop path | | 0 | |
| Weight decay | | 0.05 | |
| Data Augment | | RandomResizeAndCrop | |
| Input resolution | | $224 \times 224$ | |
| Color jitter | | 0.4 | |

Table J: Hyperparameters for fine-tuning on ImageNet-1K.

| Hyperparameters | ViT-S/16 | ViT-B/16 | ViT-L/16 |
|---|---|---|---|
| Fine-tuning epochs | 300 | 100 | 50 |
| Warmup epochs | 10 | 10 | 5 |
| Layer-wise learning rate decay | 0.6 | 0.6 | 0.8 |
| Batch size | | 1024 | |
| Adam $\epsilon$ | | 1e-8 | |
| Adam $\beta$ | | (0.9, 0.999) | |
| Base learning rate | | 5e-4 | |
| Learning rate schedule | | Cosine | |
| Repeated Aug | | ✗ | |
| Weight decay | | 0.05 | |
| Label smoothing $\varepsilon$ | | 0.1 | |
| Stoch. depth | 0.1 | 0.1 | 0.2 |
| Dropout | | ✗ | |
| Gradient clipping | | ✗ | |
| Erasing prob. | | 0.25 | |
| Input resolution | | $224 \times 224$ | |
| Rand Augment | | 9/0.5 | |
| Mixup prob. | | 0.8 | |
| Cutmix prob. | | 1.0 | |
| Relative positional embeddings | | ✓ | |
| Shared relative positional embeddings | | ✗ | |

Table K: Hyperparameters for fine-tuning on ADE20K.

| Hyperparameters | ViT-B/16 |
|---|---|
| Input resolution | $512 \times 512$ |
| Peak learning rate | 5e-5 |
| Fine-tuning steps | 160K |
| Batch size | 16 |
| Adam $\epsilon$ | 1e-8 |
| Adam $\beta$ | (0.9, 0.999) |
| Layer-wise learning rate decay | 0.75 |
| Minimal learning rate | 0 |
| Learning rate schedule | Linear |
| Warmup steps | 1500 |
| Dropout | ✗ |
| Stoch. depth | 0.1 |
| Weight decay | 0.05 |
| Relative positional embeddings | ✓ |
| Shared relative positional embeddings | ✗ |

