# OpenReview forum: "Morphing Tokens Draw Strong Masked Image Models"
_ICLR.cc/2025/Conference — ICLR 2025 Poster_

### Official Review · Reviewer_L29Y · 2024-10-31

**Soundness:** 3
**Presentation:** 3
**Contribution:** 3
**Rating:** 6
**Confidence:** 4

**Summary:**

This paper proposes Dynamic Token Morphing (DTM) to mitigate the identified spatial inconsistencies discriminative problem. This method can be used fot both masked image modeling methods and contrastive methods and accelerating the training. The achieved results look good and I think the novelty is also good. But I concern several weaknesses as I noted below. I may change the rate after reviewing other reviewers' review.

**Strengths:**

1.A interesting insight on masked image modeling pre-training methods and good results are achieved across all scale models including ViT-S/B/L.
2.The proposed method can be used for both MIM and contrastive learning mothods

**Weaknesses:**

1.Why not compare your method with hierarchical ViTs, such as ConvMAE [1], HiViT [2], iTPN [3], GreenMIM [4]. Is your method still effective on these methods? I suggest conducting an experiment on maybe one Small scale model.
2.The authors argue that the current methods meets the trouble of spatial inconsistencies, but existing ones actually have achieved very good results, such as EVA [5], EVA02 [6], Fast-iTPN [7]. How the authors expain that? If you think the proposed method can still improve their performance, I think conducting experiments is necessary.
3.Missing COCO experiments

[1]ConvMAE: Masked Convolution Meets Masked Autoencoders, [NeurIPS 2022]
[2]HiViT: A Simpler and More Efficient Design of Hierarchical Vision Transformer, [ICLR2023]
[3]Integrally Pretrained Transformer Pyramid Networks, [CVPR2023]
[4]Green Hierarchical Vision Transformer for Masked Image Modeling, [NeurIPS 2022]
[5]EVA: Exploring the Limits of Masked Visual Representation Learning at Scale [CVPR2023]
[6]EVA-02: A Visual Representation for Neon Genesis
[7]Fast-iTPN: Integrally Pre-Trained Transformer Pyramid Network with Token Migration [TPAMI2024]

**Questions:**

See the second point of Weaknesses.

---

> ### Author Response · Authors · 2024-11-21
>
> We deeply appreciate the positive reviews and valuable and insightful comments that encourage our work. We have carefully read the comment and addressed the concern through this response.
>
> > **Q1. The applicability to hierarchical ViTs**
>
> **A1.** Thank you for the comment. We did not compare with methods based on hierarchical architectures because our approach focuses on masked image modeling (MIM) pre-training, specifically handling all tokens simultaneously for Vision Transformers. In contrast, hierarchical architectures, which usually share a non-isotropic design, typically require tailored MIM techniques.
>
> However, following Reviewer L29Y's suggestion, we give some experimental results using our method DTM built upon a hierarchical architecture in a pre-training phase. Due to time constraints and the higher training costs of hierarchical ViTs (which are believed to be optimized further),  we employed the ConvMAE-Small architecture to showcase that DTM could improve performance even with fewer training epochs on ImageNet-100, pre-training and fine-tuning for 25 epochs and 100 epochs, respectively. The results in the table below demonstrate that our method when applied during pre-training can enhance the performance of a hierarchical ViT. We will provide full training results on ImageNet-1K in the revised paper.
>
>
> * Applicability of DTM on ConvMAE-Small
>     * Pre-trained for 25 epochs and fine-tuned for 100 epochs
> | Method   | Fine-tuning Accuracy (%) |
> |------------------------------|:-----------------------:|
> | ConvMAE       | 82.4                 |
> | ConvMAE + DTM | **83.0**                 |
>
>
>
>
>
>
>
>
>
>
> > **Q2. How do the authors address spatial inconsistencies when existing methods like EVA, EVA02, and Fast-iTPN have already achieved strong results?**
>
> **A2.** The suggested methods such as EVA01/02 or Fast-iTPN have shown outstanding performance. However, we hypothesize that spatial inconsistencies still exist in the frameworks due to their reliance on supervisory signals from a pre-trained teacher like CLIP or EVA-CLIP during pre-training.
>
> To investigate this, we examine the spatial inconsistency of visual token predictions generated by other supervisory models following the same approach used for CLIP in Sec 3.1, by evaluating token-wise zero-shot classification results. As suggested by Reviewer L29Y, we employ a strong and large-scale model EVA-02 and analyze its teacher model EVA-01-CLIP-g/14. We believe this analysis highlights the spatial inconsistency beyond CLIP, which may incur inaccurate supervision when used as a supervisory signal.
>
>
> We first provide an illustrative figure (please refer to **Fig A** in Appendix) which follows the same illustrative approach as Fig. 1 in the main paper. We observe that token-wise zero-shot predictions without token aggregation exhibit spatially inconsistent results compared with the zero-shot prediction results with token aggregation (it follows a trend similar to CLIP). Moreover, we observe that zero-shot accuracies computed using token-wise ensembles achieve 53.1% with token aggregation compared to 51.2% without aggregation. This highlights spatial inconsistency still exists in even a stronger pre-trained model, which can still benefit from token aggregation when used as a supervisory signal.
>
>
> Another option suggested by Reviewer L29Y, EVA-01 benefits from significantly larger training datasets (including ImageNet-21K and additional datasets) while utilizing the CLIP supervision and a BEIT framework very similar to ours. Therefore, we believe our analysis with CLIP (in the main paper), which revealed spatial inconsistency, would impact further improving EVA-01 in that it employs the same supervisory signal (CLIP).
>
>
> We plan to improve EVA01/02 using DTM during pre-training by replicating the setups detailed in their respective publications in our revised paper.

---

> ### Author Response · Authors · 2024-11-21
>
> > **Q3. Missing COCO experiments**
>
> **A3.** Thank you for suggesting the experiment. We actually reported ADE20K semantic segmentation results in Table 3 to demonstrate that our method performs effectively on a dense prediction task.
>
> Following reviewer L29Y’s suggestion, we have run some COCO experiments and report the performance improvement achieved by employing a DTM-enhanced ImageNet-pretrained backbone, which is further fine-tuned on the COCO dataset. We compare the pre-trained model using DTM with the baseline model that does not incorporate DTM. We employ Mask R-CNN with ViT-B/16 and initialize its encoder with the baseline and our DTM models, respectively. Due to time constraints and the high computational cost associated with high image resolutions on COCO, we adopted a lighter experimental setup than the standard training protocols used in previous self-supervised learning methods. We resized the larger side of input images to 1024 and fine-tuned the models for 10 epochs; we further reduced the image resolution to 512 and fine-tuned the models for 18 epochs as another experimental setup. We observe that DTM enhances the baseline, delivering consistent improvements across all the metrics in both experimental setups.
>
> We again appreciate your understanding that we employed lighter experimental setups due to time and resource constraints, training with fewer epochs (upper table) and smaller resolution (lower table). We observe that the backbones pre-trained with our method achieve significant gains across both experimental setups. We will include experiments with the standard experimental setups using larger models in the revised paper.
>
> * Mask RCNN fine-tuning (res: 1024, trained for 10 epochs)
> | Method           | bbox/AP | bbox/AP50 | bbox/AP75 | bbox/APs | bbox/APm | bbox/API | segm/AP | segm/AP50 | segm/AP75 | segm/APs | segm/APm | segm/API |
> |------------------------------|---------|-----------|-----------|----------|----------|----------|---------|-----------|-----------|----------|----------|----------|
> | Baseline                    | 46.7    | 69.4      | 51.1      | 29.6     | 50.5     | 61.6     | 41.9    | 65.6      | 45.0      | 22.2     | 44.7     | 61.3     |
> | DTM                         | **47.4**    | 69.8      | 51.9      | 30.6     | 51.3     | 62.3     | **42.5**    | 66.6      | 45.6      | 22.6     | 45.7     | 61.3     |
> | Gain                | 0.7     | 0.4       | 0.8       | 1.0      | 0.8      | 0.7      | 0.6     | 1.0       | 0.6       | 0.4      | 1.0      | 0.0      |
>
>
> * Mask RCNN fine-tuning (res:512, trained for 18 epochs)
> | Method | bbox/AP | bbox/AP50 | bbox/AP75 | bbox/APs | bbox/APm | bbox/API | segm/AP | segm/AP50 | segm/AP75 | segm/APs | segm/APm | segm/API |
> |------------------------------|---------|-----------|-----------|----------|----------|----------|---------|-----------|-----------|----------|----------|----------|
> | Baseline                    | 41.8    | 63.9      | 45.3      | 21.1     | 46.1     | 61.4     | 37.5    | 60.1      | 39.7      | 15.0     | 40.2     | 61.0     |
> | DTM                         | **42.9**    | 65.0      | 46.3      | 22.2     | 47.5     | 63.0     | **38.2**    | 61.0      | 40.6      | 16.1     | 41.1     | 61.8     |
> | Gain                 | 1.1     | 1.1       | 1.0       | 1.1      | 1.4      | 1.6      | 0.7     | 0.9       | 0.9       | 1.1      | 0.9      | 0.8      |

---

> ### Author Response · Authors · 2024-11-25
> **A gentle reminder to look over our responses**
>
> Dear Reviewer L29Y
>
> We sincerely appreciate your thoughtful positive review and constructive comments, which greatly encouraged our work. As the discussion period nears its end, we would like to leave a gentle reminder to consider our responses. We have made every effort to address all your concerns by including requested experimental results and detailed explanations (along with paper revisions). Please feel free to let us know if you have any additional questions or require further clarification.
>
> Thank you once again for your valuable reviews.
>
>
> Best regards,
>
> Authors

---

> > ### Comment · Reviewer_L29Y · 2024-11-26
> > **Thanks**
> >
> > I think all my concerns all addressed. Thanks for the responses.
> >
> >
> > Considering the outstanding performance of this paper, I choose to weakly accept it. However, I want to emphasize that I believe this paper will not bring significant changes to the MIM field, as this area will likely continue to be dominated by the simplest pretraining approaches using CLIP as a supervisory signal (e.g., EVA, EVA02, fast-itpn). Therefore, if the AC decides to reject this paper, I would also find it acceptable.

---

> ### Author Response · Authors · 2024-11-27
> **Reply to Reviewer L29Y**
>
> Dear Reviewer L29Y
>
>
> We sincerely appreciate your response and consideration of our paper as a weak acceptance.
>
>
> In the current era, extensively utilizes CLIP-supervisory signals, we believe our method would offer broader applicability by effectively enhancing the signals in that the effectiveness has already been demonstrated in various ways. In our paper, we showcased the general applicability of DTM across several architectural variations (i.e., MAE+CLIP, BEiT-v2, and BYOL) and a variation of CLIP (i.e., SLIP). During the rebuttal period, we further investigated the applicability of DTM to EVA CLIP, validating its general effectiveness on variations of CLIP.
>
>
> Besides the general applicability of DTM, recent Masked Image Modeling (MIM) studies have mainly focused on innovations in framework design and improved masking strategies. This remains significant potential for refining their MIM supervision using CLIP or other signals. From this perspective, we believe our work can become a baseline to improve MIM supervision through an intuitive plug-and-play manner, with the potential to inspire future research in this direction.
>
>
> Finally, we sincerely thank you again for your positive feedback and greatly welcome any additional concerns or feedback required for more favorable consideration.
>
>
> Best regards,
>
> Authors

---

### Official Review · Reviewer_gb6h · 2024-11-06

**Soundness:** 2
**Presentation:** 2
**Contribution:** 3
**Rating:** 6
**Confidence:** 4

**Summary:**

This paper investigates the issue of spatially inconsistent target representations during pre-training. The authors conducted quantitative and qualitative experimental observations of this phenomenon to discuss the impact of such spatial inconsistencies, particularly on the performance of downstream classification tasks. To address this issue, based on the MIM baseline, they propose an advanced token aggregation method termed Dynamic Token Morphing (DTM), which integrates tokens with contextual associations. The paper reports the classification results on ImageNet-1k and the segmentation performance on ADE20k, demonstrating consistent improvements with DTM across the SSL framework and the SLIP model.

**Strengths:**

1.This paper provided an extensive discussion and demonstration of the spatial inconsistency in pre-trained representations, illustrating the significance of this issue in the domain of self-supervised learning.
2. DTMs re-characterize visual features, mitigating the inflexibility associated with patch-based representations.
3. This paper integrated DTM with multiple baseline methods, such as MAE, BEiT v2, BYOL, etc., demonstrating the effectiveness and convenience of the proposed approach.

**Weaknesses:**

1. Although DTM presents an intuitive feasibility in addressing the spatial inconsistency of visual representations, the explanation in the methods section is not very intuitive or detailed, which may lead to comprehension errors. For instance, the process of how to perform k iterations based on bipartite matching to obtain the final token morphing matrix M, and how to derive the aggregated results from M, is not clear. Perhaps an algorithm flowchart would be beneficial.
2. Some details in the writing of the article could be further optimized. For example, on line 264, whether generating vi with xiM should not include masked tokens, as there is an inconsistency with Figure 4 (the input of the target encoder does not include masked tokens). Additionally, in Figure 4, the lines connecting the Morphing Matrix to Vi and Ui are different; does this indicate different operations?
3. The authors emphasize in the abstract that DTM brings about faster training, but when comparing with other methods, especially the baseline method, the best results are achieved using the same number of epochs. Does this imply that the overall training time was reduced?

**Questions:**

In addition to weakness, there are also some questions:
1. In line 239, "we do not employ a superpixel-based clustering for our method, which is inefficient as well." Is the decision to avoid superpixel-based clustering solely due to its low performance? Could the reasons be analyzed further?
2. Please explain why the absence of a dynamic mechanism results in lower performance than the baseline in Table 7.

---

> ### Author Response · Authors · 2024-11-21
>
> We deeply appreciate the detailed reviews and valuable and insightful comments that encourage our work. We have carefully read the comment and addressed the concern through this response.
>
> > **Q1. Enhancing clarity and details in Section 4.2**
>
> **A1.** We apologize for the lack of detailed descriptions addressing the Reviewer gb6h's concerns. Following the advice, we have improved the manuscript and included an algorithm flow table to explain the morphing matrix generation process in greater detail (please refer to **Section 4.2** and  **Section B** in Appendix). The revised texts have been marked in blue for clarity. The key updates are summarized as follows:
> We have added a detailed description of the process of performing k iterations with bipartite matching;
> We revised to clarify how the aggregated results are derived from the token morphing matrix M;
> An algorithm flowchart for the token morphing function, which generates M,  has been added.
>
> > **Q2. More refined writings**
>
> **A2.** We first apologize for the confusion. Following the Reviewer gb6h's suggestion, we have provided additional clarifications.
> As Reviewer gb6h commented, in line 264, $\textbf{v}_ i=f_\xi(x^M_i)$ should be $\textbf{v}_ i=f_\xi(x_i)$.
> The line connections appear inconsistent but should be corrected to have the same shape. We apologize for the confusion. Specifically, the lines from the morphing matrix M to each side now have arrows, which denote the morphed tokens ($\textbf{v} → \hat{\textbf{v}}$) and ($\textbf{u} → \hat{\textbf{u}}$) generated by the same morphing operation. In the revised paper, we have corrected the lines in **Fig.4** to clarify that the morphing matrix is applied identically to all tokens both $\\{ \mathbf{u}_ {i} \\}^N_{i=1}$ and $\\{ \mathbf{v}_ {i} \\}^N_{i=1}$.
> We hope our revisions provide greater clarity and address any previously misleading details.
>
>
>
> > **Q3. On faster training and overall training time**
>
> **A3.** Faster training refers to a faster convergence of the training metrics during fine-tuning. As shown in Fig. 5, our approach reaches a given accuracy faster than the baseline and achieves lower loss values earlier, which demonstrates a faster convergence from our perspective. Using the same number of epochs is to remain consistent with the baseline and for a fair comparison. Since our method reaches the same accuracy more quickly, extending the training to larger epochs provides additional performance benefits, as also observed in Fig. 6.
>
> Additionally, regarding training time, Table 6 compares DTM's training speed with the baseline. While DTM introduces minor additional computational costs due to token morphing and alignment loss, its impact on throughput is minimal (<1%) and yields significant performance improvements. As a result, DTM maintains a training speed very similar to the baseline, though the overall training time may not be reduced.
>
> > **Q4. Explanation for not using superpixel-based clustering**
>
> **A4.** The main reason for not using superpixel-based clustering was its inefficiency, despite a slightly improved distillation performance shown in Table 2. While the results in the table suggest the potential of context-aware aggregation methods, we argue their effectiveness should be validated through a more practical MIM-focused study that aligns more closely with our goal. Specifically, as noted in lines 237-238, our pilot experiment found that learning with superpixel-based clustering achieved a baseline improvement of 87.1%; however, employing alternative - Bipartite Matching (used in DTM) - yielded a higher performance of 87.9% (over the baseline 79.5%). We attribute the suboptimal result of the superpixel-based clustering to two factors: 1) its reliance on a fixed number of clustering centers and 2) its limitation of generating a single set of morphed tokens.
>
>
> > **Q5. On the lower performance without the dynamic mechanism in Table 7**
>
> **A5.** Without a dynamic approach, Table 7 exhibits that token aggregation uses a fixed number of tokens in a single stage, which can lead to performance degradation. This occurs because determining the optimal number of tokens is challenging—morphing too many tokens erases detailed information while morphing too few experiences a limited impact. Additionally, performing morphing in a single step may reduce token diversity and introduce noise due to the inherent randomness of the matching algorithm (e.g., random splits in Bipartite Matching). This can result in misaligned tokens which would restrict the model's capability to learn more localizable representations. The observed performance drop highlights how undesirable, non-dynamic morphing affects representation quality; we believe this result emphasizes the need for the dynamic mechanism in DTM.

---

> > ### Comment · Reviewer_gb6h · 2024-11-28
> > **Thanks**
> >
> > Thanks for the responses, which has basically answered my question. I am willing to raise my score, but I reserve the following opinion: It is generally the case that when most algorithms outperform the baseline, it is accompanied by faster convergence of the loss, making it difficult to be considered a significant contribution of this work.

---

> > > ### Author Response · Authors · 2024-11-28
> > > **Reply to Reviewer gb6h**
> > >
> > > Dear Reviewer gb6h
> > >
> > > We sincerely appreciate the positive re-rating, which has greatly encouraged us to further improve our work. We also agree with reviewer gb6h’s observation that effective methods generally tend to exhibit faster training. While our method was not explicitly designed to specifically target faster training, our observations suggest that enhancing the supervisory signals with our approach improves spatial consistency, which in turn seems to accelerate convergence (as shown in Fig. 6). Furthermore, our method speeds up training while lowering final loss under a same training setup, which we believe offers another minimal contribution beyond its primary strengths: 1) addressing spatial inconsistency, 2) broad applicability, and 3) performance improvements.
> > >
> > >
> > > Finally, we sincerely thank you again for the positive rating and for the insightful and thorough reviews. We warmly welcome any additional feedback or suggestions to further improve our paper.
> > >
> > >
> > > Best regards,
> > >
> > > Authors

---

> ### Author Response · Authors · 2024-11-25
> **A gentle reminder to look over our responses**
>
> Dear Reviewer gb6h
>
> We sincerely appreciate your thoughtful review and constructive comments, which have greatly encouraged our work. As the discussion period nears its end, we would like to leave a gentle reminder to consider our responses. We have made every effort to address all your concerns by providing detailed explanations and corresponding paper revisions. Please feel free to let us know if you have any additional questions or require further clarification.
>
> Thank you once again for your valuable reviews.
>
>
> Best regards,
>
> Authors

---

### Official Review · Reviewer_xEMo · 2024-11-08

**Soundness:** 4
**Presentation:** 4
**Contribution:** 3
**Rating:** 8
**Confidence:** 3

**Summary:**

The paper proposes a novel masked image modeling method. The paper identified the issue of the existing pre-training models: the spatial consistency in target representations despite proximity and contextual similarity. Then the author proposed a context-aware token aggregation method called Dynamic Token Morphing (DTM). The experiments show that the proposed DTM method is general enough to help improve various SSL frameworks and achieve SOTA ImageNet-1k performance. In addition, the computation complexity add-on of the proposed method is not large.

**Strengths:**

- The paper achieves SOTA results on ImageNet-1k.
- The proposed method is applicable to various SSL frameworks such as MAE, BEiT and BYOL.
- The paper is very well written and has extensive experiments and visualizations.
- The "Pilot Study" section gives very good narrative to the paper and provides intuitive motivations.

**Weaknesses:**

The paper is very well-written and achieves SOTA results, I don't have weakness to add.

**Questions:**

- For the Token encoding section of "4.1 Preliminary" section, consider adding more introduction/references to the concept introduced, for example "online encoder" and "target encoder.
- It will be easier for readers to follow if there is an additional figure about how to obtain the morphing matrix M.

---

> ### Author Response · Authors · 2024-11-21
>
> We sincerely appreciate the positive reviews and the valuable, insightful comments encouraging our work. We have carefully considered the feedback and addressed the concerns in this response.
>
> > **Q1.More introductions and references to the concept for clarity**
>
> **A1.** Thank you for the advice. We have complemented the explanation of the introduced concepts and revised some explanations. In “4.1 Preliminary", we have revised the manuscript according to the Reviewer xEMo's suggestion as follows:
> * We have added references [A, B] to support the block-wise masking approach;
> * We have clarified some missing descriptions of the concepts we introduced such as the inputs and outputs of the online and target encoders, as well as the definitions of the terms themselves.
>     * The terms "online encoder" and "target encoder" are derived from existing literature [C, D, E]. We have updated the manuscript to include the description.
>
> We hope these revisions improve clarity and address the concerns.
>
> [A] BEiT: BERT Pre-Training of Image Transformers, ICLR 2022
>
> [B] Beit v2: Masked image modeling with vector-quantized visual tokenizers, arxiv 2022
>
> [C] Bootstrap your own latent: A new approach to self-supervised Learning, NeurIPS 2020
>
> [D] iBOT: Image BERT Pre-Training with Online Tokenizer, ICLR 2022
>
> [E] data2vec: A General Framework for Self-supervised Learning in Speech, Vision and Language, ICML 2022
>
>
> > **Q2.Request for the illustrative figures of the morphing matrix M**
>
> **A2.**
> - We have added an illustrative figure in the methods section (please refer to **Fig.5** in Section. 4) to illustrate the overall process of obtaining the morphing matrix M, in response to the comment.
>
> - Additionally, we have included an algorithm flowchart for the Token Morphing Function (in lines 864-881), which details the iterative process of generating the morphing matrix M.

---

> ### Author Response · Authors · 2024-11-25
> **A gentle reminder to look over our responses**
>
> Dear Reviewer xEMo
>
>
> We sincerely appreciate your thoughtful and positive review, as well as constructive comments, which have significantly encouraged our work. As the discussion period nears its end, we would like to leave a gentle reminder to consider our responses. We have made every effort to address all your concerns by incorporating the requested paper revisions and providing detailed explanations. Please feel free to let us know if you have any additional questions or require further clarification.
>
> Thank you once again for your valuable reviews.
>
> Best regards,
>
> Authors

---

> > ### Comment · Reviewer_xEMo · 2024-11-27
> > **Thank you**
> >
> > Thank you addressing the comments. The added Fig.5 in Section. 4 is very helpful for understanding. I will maintain my scores.

---

> ### Author Response · Authors · 2024-11-28
> **Reply to Reviewer xEMo**
>
> Dear Reviewer xEMo
>
> We greatly appreciate your positive feedback and are glad to hear that our responses have addressed your concerns. We welcome any additional feedback or suggestions for further improvement of our paper.
>
> Best regards,
>
> Authors

---

### Author Response · Authors · 2024-11-21
**General Response**

We thank all the **Reviewers xEMo, gb6h, L29Y** for their thorough and constructive reviews with valuable and insightful advice. We appreciate the positive comments from all reviews:

**Reviewer xEMo** - 1) our **state-of-the-art (SOTA) results** on ImageNet-1K, 2) **wide applicability** to masked image modeling (MIM) and self-supervised learning (SSL) methods, 3) **high clarity** of the manuscript, 4) **extensive experiments/visualizations**, and 5) the **insightful motivation** and **very good narrative to the paper provided in the pilot study**;

**Reviewer gb6h** - 1) **extensive discussions/demonstrations** of spatial inconsistency in our work that highlight our method’s significance, 2) **effective solution** addressing a novel problem, and 3) **broad applicability** to both MIM and SSL;

**Reviewer L29Y** - 1) **interesting insights** on MIM pre-training, 2) **strong results** achieved across all model scales including ViT-S/B/L, and 3) **wide applicability** of our method to MIM and SSL frameworks.

We have made every effort to address all the comments by carefully revising the paper. Throughout the revised paper, we highlighted the newly added or edited materials **in blue**. Please refer to the following summarization of the revised materials as follows:
* Complemented the explanation of the introduced concepts and revised some explanations with references in “4,1 Preliminary” (**Q1** by **Reviewer xEMo**)
* Added an illustrative figure of the morphing matrix M in Fig. 5 (**Q2** by **Reviewer xEMo**)
* Provided additional clarifications in Section 4.2 (**Q1** by **Reviewer gb6h**)
* Added an algorithm flowchart (in Section B of Appendix) to detail the flow of the Token Morphing Function (**Q1** by **Reviewer gb6h**, **Q2** by **Reviewer xEMo**)
* More refined writings in Section 4.1 and Fig. 4 (**Q2** by **Reviewer gb6h**)
* Added visualization of the spatial inconsistency of EVA-CLIP-g/14 in Fig. A in Appendix with a quantitative comparison in its caption (**Q2** by **Reviewer L29Y**)

Moreover, alongside our revisions, we tried our best to resolve the reviewer’s comments:
* Provided a detailed clarification of the “faster training” property (**Q3** by **Reviewer gb6h**)
* Provided an explanation for not using superpixel-based clustering (**Q4** by **Reviewer gb6h**)
* Clarified the reason for the lower performance without the dynamic mechanism in Table 7 (**Q5** by **Reviewer gb6h**)
* Showcased the effectiveness of DTM on hierarchical ViTs (**Q1** by **Reviewer L29Y**)
* Added new COCO detection results to show the effectiveness of DTM (**Q3** by **Reviewer L29Y**)

Detailed responses to other comments are left in individual comments. Due to the page limit, we send "Applicability on other targets" and "Applicability on SSL frameworks" to Section C in Appendix.

---

### Meta-Review · Area_Chair_s4nK · 2024-12-17

**Metareview:**

This paper presents an algorithm for visual pre-training. It was based on masked image modeling, and the method was motivated by the so-called spatial inconsistency between image tokens. To alleviate it, the proposed method involves merging (morphing) visual tokens into larger units according to inter-token similarity, and the morphed tokens are used as basic units in the masked image modeling procedure.

Overall, the method is interesting and shows good practice. All three reviewers are positive on this submission, making it a case of clear acceptance. However, the AC still has a few concerns about the submission.

* First, the paper does not clearly connect the proposed method with spatial inconsistency; I was not convinced why grouping similar visual tokens can solve the inconsistency issue, although qualitative experiments (e.g. the example in Figure 1) seem to suggest that such trained models enjoy better spatial consistency.
* Upon this, I am a bit doubtful about why the method works -- does it work by alleviating the inconsistency issue, or by finding a more proper difficulty for the MIM task (note that grouping visual tokens can decrease the difficulty and make some "unpredictable" units predictable)? The paper lacks sufficient analysis and discussion on this point.
* The writing of this paper needs major improvement; the current form makes it difficult to understand many things. Adding some figures in the rebuttal makes things better, but the AC still had a difficult time reading the paper. Reviewers also pointed out this issue.
* In addition, I do not think the experiments are sufficient to fully validate the effectiveness -- there are only limited ImageNet tests with some transfer tests on ADE20K and FGVC sets, and the ablations are minimal. By the way, I agree with Reviewer **gb6h** that convergence (shown in Figure 6) cannot be solid evidence.

The AC discussed this case with the SAC in length. With three positive reviews (6/6/8), a final decision of acceptance was made, but the above concerns do exist. The AC strongly suggests the authors to take them into consideration and try to improve the final paper.

**Additional Comments On Reviewer Discussion:**

The original scores of this paper were 5/6/8. After the rebuttal, the scores became 6/6/8. Two reviewers suggesting weak acceptance said that they would not defend the acceptance, and the reviewer suggesting clear acceptance seems not to have provided much information in the review and response. Therefore, the AC believes that this is still a borderline case.

---

### Decision · Program_Chairs · 2025-01-22

Accept (Poster)